# Actomyosin contractility-dependent matrix stretch and recoil induces rapid cell migration

William Y. Wang [1], Christopher D. Davidson [1], Daphne Lin[1] & Brendon M. Baker [1]

Cells select from a diverse repertoire of migration strategies. Recent developments in tunable biomaterials have helped identify how extracellular matrix properties influence migration, however, many settings lack the fibrous architecture characteristic of native tissues. To investigate migration in fibrous contexts, we independently varied the alignment and stiffness of synthetic 3D fiber matrices and identified two phenotypically distinct migration modes. In contrast to stiff matrices where cells migrated continuously in a traditional mesenchymal fashion, cells in deformable matrices stretched matrix fibers to store elastic energy; subsequent adhesion failure triggered sudden matrix recoil and rapid cell translocation. Across a variety of cell types, traction force measurements revealed a relationship between cell contractility and the matrix stiffness where this migration mode occurred optimally. Given the prevalence of fibrous tissues, an understanding of how matrix structure and mechanics influences migration could improve strategies to recruit repair cells to wound sites or inhibit cancer metastasis.

[1] Department of Biomedical Engineering, University of Michigan, Ann Arbor, MI 48109, USA. Correspondence and requests for materials should be addressed to B.M.B. (email: bambren@umich.edu)

Cell migration, a fundamental biological process in embryogenesis, tissue homeostasis, and cancer metastasis, involves dynamic interactions between cells and their local microenvironment[1,2]. Biochemical and biophysical characteristics of the surrounding extracellular matrix (ECM) influences cell migration through variations in growth factors or chemokines (chemotaxis), stiffness (durotaxis), ligand density (haptotaxis), and topographical organization (contact guidance) to direct cells to target destinations[3]. Recent advances in intravital imaging have revealed that cells can adopt a diverse set of migration strategies involving migration as single cells or collective strands, transitions between mesenchymal, epithelial, and amoeboid migration modes, deformation of the cell body and nucleus to squeeze through matrix pores, and remodeling of matrix structure to bypass the physical barriers presented by the ECM[4–6]. However, poor control over biochemical and mechanical properties of native tissues has hampered mechanistic understanding of how cells interpret and convert these external cues into the coordinated molecular signals that orchestrate cell migration. Thus, in vitro models of cell migration have proven indispensable in complementing in vivo studies to elucidate how specific ECM properties impact cell migration.

In particular, advances in tunable biomaterials and microfabricated in vitro models have helped elucidate how cells select from a repertoire of migration strategies[2,7,8]. In proteolysis-dependent migration, where cells are capable of biochemically remodeling the surrounding microenvironment to generate space to move, the degree of ECM degradability influences whether cells migrate as collective multicellular strands or escape as single cells[9,10]. Initial leader cells have been shown to use proteolytic machinery to generate microchannels within the ECM, enabling proteolysis-independent migration of follower cells[11,12]. Alternatively, cells are capable of employing a water permeation-based migration mode within microchannels[13]. In purely non-proteolytic migration, cells alter their morphology to squeeze through small ECM pores, leading to nuclear rupture and ESCRT III-mediated repair[14] or can transition between mesenchymal and amoeboid migration modes via alterations in matrix adhesivity and confinement[15]. These studies reducing the complex physical properties of native tissues to sets of orthogonally tunable parameters have not only increased our mechanistic understanding of cell migration but also identified diverse non-proteolytic migration strategies, which may in part explain the failure of therapeutics solely targeting proteolytic activity toward confining metastatic cells to the primary tumor[16].

Within microenvironments in which cells can neither modify their morphology nor proteolytically degrade the ECM to effectively migrate, cell force-mediated reorganization of physical structures of the surrounding ECM may facilitate cell movement. Fibrils in collagen and fibrin gels deform as cells apply traction forces during migration[17,18], however, poor control over mechanical properties and the inability to remove proteolysis-mediated remodeling of naturally derived ECM proteins has hampered our understanding of how physical reorganization of ECM fibrils influences migration[7,19]. Modeling the ECM with synthetic hydrogels composed of non-proteolytically cleavable crosslinks has elucidated how cells deform the ECM during migration in soft three-dimensional (3D) polyethylene glycol (PEG) hydrogels[20], however, these materials lack the fibrous architecture inherent to many native tissues[21]. For example, the fibrous matrix of the surrounding tumor stroma of breast and pancreatic cancers undergoes marked remodeling, with increases in fibril alignment and tissue stiffness as the cancer becomes progressively more metastatic[22,23]. The importance of these physical changes is underscored by their clinical use as individual prognosticators of cancer patient survival rates[24].

Toward understanding how aspects of the ECM influence dynamic interactions between cells and their physical microenvironment, here we implement a recently established synthetic material system that models fibrous ECMs and enables independent control over alignment and stiffness[25]. Examining the migration of single mesenchymal cells, we find that fiber alignment enhances migration speed and directionality, while stiffness elicits a biphasic response with a maximum migration speed occurring at an intermediate matrix stiffness. Interestingly, cells within deformable matrices adopt a unique migration phenotype where cell contractility-generated matrix stretch and subsequent recoil result in rapid migratory events with effective speeds >5× than previously reported. We term this mode slingshot migration (SSM) given the requirement for matrix stretch and recoil, and further demonstrate that SSM events occur most frequently at an intermediate matrix stiffness, in part contributing to the biphasic response of migration speed to matrix stiffness. Lastly, we find that a variety of mesenchymal cell types employ this migration mode and baseline cell contractility determined by traction force microscopy (TFM) correlates with the optimal matrix stiffness where this migration mode optimally occurs.

## Results

**Synthetic matrices with tunable fiber alignment and stiffness.** To better understand how matrix alignment and stiffness of fibrous ECM influence cell migration, we designed and characterized a synthetic ECM mimetic composed of electrospun dextran methacrylate (DexMA) fibers with orthogonal control over fiber alignment and bulk stiffness[25]. Cell-perceived ECM mechanics were controlled by electrospinning fibrous matrices over an array of microfabricated poly(dimethylsiloxane) (PDMS) wells (Fig. 1a), such that cells seeded in well regions are not influenced by a mechanically rigid underlying support layer. To modulate fiber alignment, we altered the shape of the electric field at the collecting surface during electrospinning by controlling the separation distance between two parallel collecting electrodes[26], where increasing this distance increased fiber alignment (Fig. 1b, c). We tuned photo-initiated DexMA crosslinking via ultraviolet (UV) exposure to modulate stiffness at the single fiber (measured by three-point bending with atomic force microscopy (AFM)) and bulk substrate level (measured in tension by indentation of suspended matrices with a 1 mm cylindrical indenter) (Fig. 1d, Supplementary Fig. 1a), selecting a range of crosslinking to capture a full spectrum from maximal to undetectable matrix deformations resulting from cell-generated forces (Fig. 1e). These values for single fiber Young's modulus are within range of reported values for fibrin fibers, elastin fibers, and fibronectin fibrils (depending on their stretch state)[27]. The range for bulk stiffness values (1–30 kPa) reflects measurements taken for a variety of tissues[28], as well as the transition from normal to cancerous mammary tissue[29]. We held ligand density, fiber diameter, and scaffold thickness constant by maintaining a fixed concentration of cRGD, polymer weight percentage, and fiber collection duration, respectively (Supplementary Fig. 1b–e). Controlling these matrix properties within naturally derived ECM proteins such as collagen and fibrin hydrogels proves challenging, as these properties are inherently linked to protein concentration and gelation conditions[7,19,30]. By creating matrices with free-radical polymerized DexMA fibers, which generates proteolytically uncleavable methacrylate-methacrylate crosslinks, we side-stepped the influence of cell-mediated matrix degradation on matrix properties over time[10,31,32] and focused here on non-proteolytic mechanisms of 3D cell migration[33].

Utilizing NIH3T3 fibroblasts, commonly employed in studying how microenvironmental cues govern mesenchymal cell

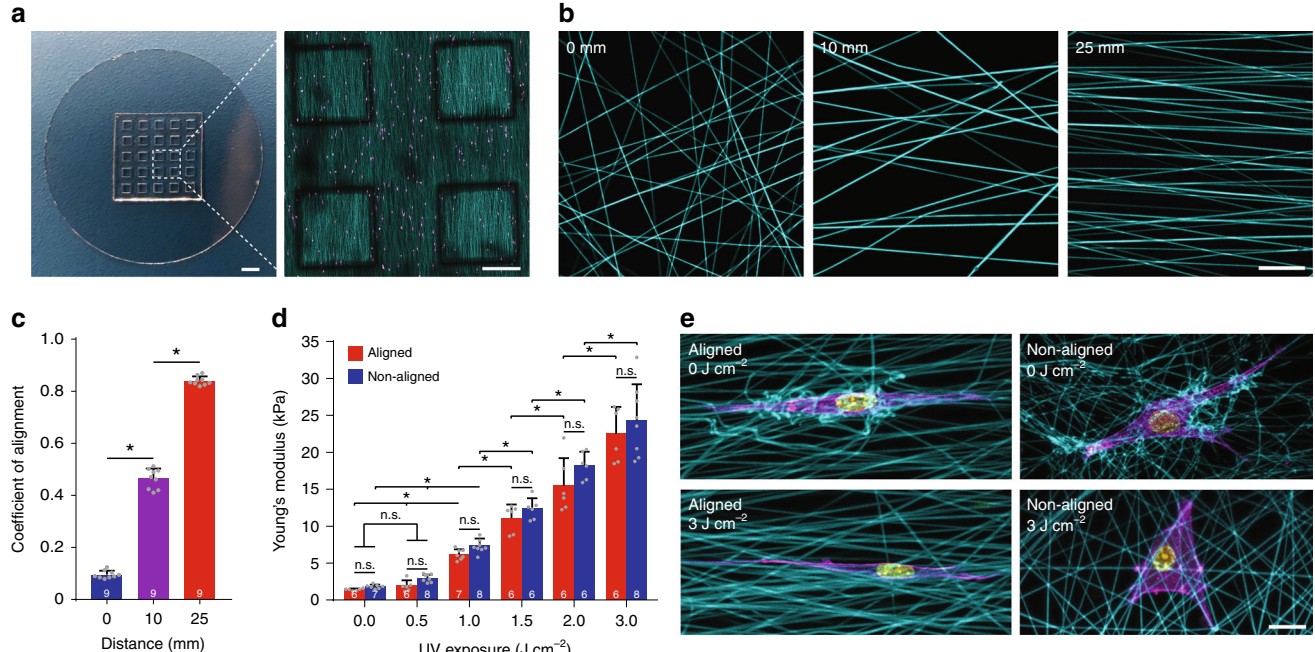

**Fig. 1** Synthetic fibrous extracellular matrix with orthogonal control over fibril alignment and stiffness. **a** Microfabricated poly(dimethylsiloxane) multiwell substrate possessing a 5 × 5 array of DexMA fiber matrices, each suspended over a well to isolate the matrix from mechanical effects of a rigid underlying support (scale bar: 2 mm). Inset: tilescan of four DexMA-cRGD matrices (cyan) seeded with NIH3T3 fibroblasts (magenta) (scale bar: 500 μm). **b** Confocal projections of matrices (cyan) fabricated with varying spacing between collecting electrodes to modulate fiber alignment (scale bar: 20 μm). **c** Quantification of fiber alignment as a function of electrode separation distance (n = 9 matrices per group). **d** Young's modulus of aligned (electrode separation of 0 mm) and non-aligned (electrode separation of 25 mm) matrices as a function of ultraviolet (UV)-initiated crosslinking of matrix fibers. n = number of matrices per group as indicated within each bar. **e** Composite confocal fluorescence images of representative NIH3T3 fibroblasts in soft (top row, 0 J cm$^{-2}$) and stiff (bottom row, 3 J cm$^{-2}$) matrices; rhodamine-labeled matrix fibers (cyan), F-actin (magenta), and nuclei (yellow) (scale bar: 20 μm). All data presented as mean ± s.d.; * indicates a statistically significant comparison with p < 0.05; n.s. indicates a non-significant comparison (one-way analysis of variance)

migration[34,35], we first investigated the effect of fiber alignment at three distinct stiffness conditions (for simplicity, referred to throughout the text as low (1.42 kPa), intermediate (6.17 kPa), and high (22.5 kPa) stiffness corresponding to 0, 1, and 3 J cm$^{-2}$ of UV exposure, respectively). Cells were cultured for 6 h prior to the start of time-lapse imaging, resulting in the majority of cells infiltrating and embedding themselves within the matrix (Supplementary Fig. 1f). At each level of stiffness, aligned matrices resulted in higher migration speeds and more directional migration tracks (Fig. 2a–d, Supplementary Movie 1). The effect of alignment on migration directionality (determined by the deviation of a cell's position away from a linear fit to its overall migration track) proved consistent over all stiffnesses, suggesting the influence of contact guidance[35,36] on directionality is stiffness-independent in this setting. This finding is consistent with previous studies utilizing 3D collagen hydrogels[30].

Within aligned matrices, these initial studies indicated migration speeds were highest at an intermediate stiffness (Fig. 2b). We next modulated matrix stiffness over more graded steps, confirming a biphasic relationship between migration speed and bulk matrix stiffness for NIH3T3s (Fig. 2e, Supplementary Fig. 2a, b). Demonstrating this response is not unique to NIH3T3s, we tracked human foreskin fibroblast migration within aligned matrices of varying stiffness and similarly found a biphasic response between migration speed and bulk matrix stiffness (Supplementary Fig. 2c). Such a biphasic relationship between migration speed and stiffness has been previously predicted by mathematical models and supported with experimental work on two-dimensional (2D) hydrogels[37–43]. These studies indicate matrix properties that optimize integrin

engagement and traction generation feed progression of the cell migration cycle, resulting in optimal migration speeds. In particular, high matrix stiffness or ligand density leads to stable focal adhesions preventing detachment of the cell rear, and low matrix stiffness or ligand density can inhibit cell contractility and cell-generated traction forces required to contract the cell body forward. However, previous studies did not examine the effect of cell-scale local matrix deformations, which potentially could exert a pronounced influence on cell motion in soft fibrous settings[25].

**SSM contributes to biphasic migration speed.** Toward understanding one possible contribution to the biphasic response between migration speed and stiffness within aligned matrices, we imaged at higher spatiotemporal resolution to observe dynamic cell-matrix interactions during migration. Cells were observed to adopt two distinct migration modes as a function of matrix stiffness. Across all stiffnesses examined, cells appeared to migrate using the well-described mesenchymal cell migration cycle consisting of iterative rounds of elongation, adhesion, contraction, and retraction[44] (henceforth termed continuous migration) (Fig. 3a, c–e). However, in deformable matrices, cells were also observed to undergo a prolonged extension/contraction phase, during which the cell's position remained largely stagnant while active cell protrusions reorganized the adjacent matrix by recruiting fibers toward the cell body. Following this phase of matrix deformation, an apparent failure in adhesions at the cell's trailing edge led to a sudden recoil of the matrix, simultaneous with translation of the cell body forward along the axis of fiber alignment (Fig. 3b, f; Supplementary Movie 2). Given the

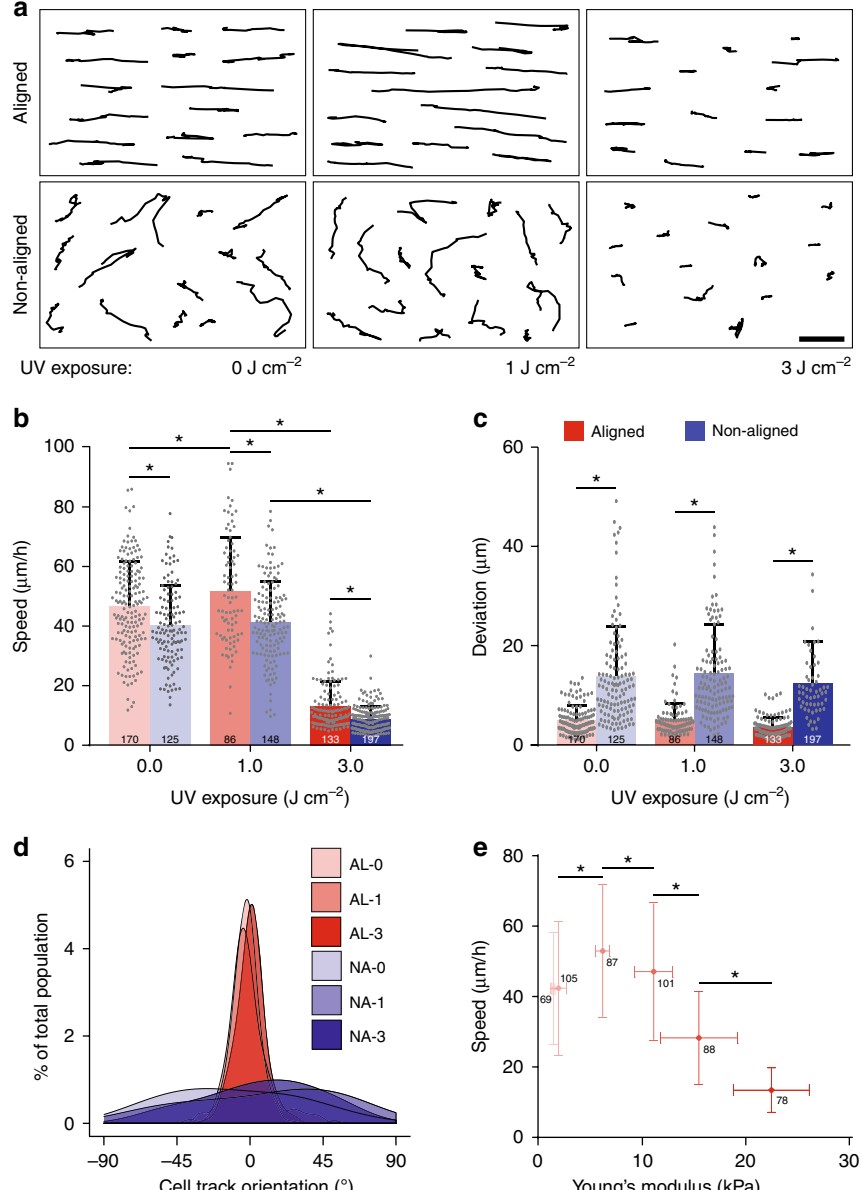

**Fig. 2** Matrix alignment and stiffness influence cell migration speed and directionality. **a** Representative NIH3T3 migration tracks over a 6 h time course on aligned and non-aligned matrices at low ($0 \, J \, cm^{-2}$), intermediate ($1.0 \, J \, cm^{-2}$), and high ($3.0 \, J \, cm^{-2}$) ultraviolet (UV) exposure; scale bar: 100 μm. **b–d** NIH3T3 net migration speed, deviation, and orientation as a function of matrix alignment and UV exposure. $n =$ number of cells per group as indicated within each bar. **e** Migration speed as a function of Young's modulus in aligned matrices. $n =$ number of cells per group as indicated beside each data point. All data presented as mean ± s.d.; * indicates a statistically significant comparison with $p < 0.05$ (two-way analysis of variance)

phenotypic departure from traditional continuous migration, we termed this mode SSM, as the cell appears to harness matrix stretch and recoil to slingshot forward. At an intermediate stiffness, we found the percentage of the total cell population that employed SSM (SSM population, 71.2%) and the percentage of tracked time cells underwent SSM (SSM duration, 31.6%) were highest (Fig. 3h, i), as high stiffness matrices appeared insufficiently deformable to afford appreciable matrix stretch while low stiffness matrices, although significantly stretched, appeared too compliant to consistently induce a recoil event (Supplementary Movie 2).

Comparing migration distance over time of these two distinct modes, we observed steady migration speeds with continuous migration, whereas SSM consisted of two distinct speeds corresponding to separate phases of matrix stretch and recoil (Fig. 3g). Furthermore, cells within deformable matrices (both soft

and intermediate stiffness) interconverted between continuous and SSM modes over the course of their tracked lifetimes (Fig. 3j). Of cells that underwent SSM within intermediate stiffness matrices, individual cell trajectories parsed into periods of continuous migration and SSM (possessing distinct phases of stretch followed by recoil) revealed that the large recoil distance ($58.7 \pm 21.3 \, \mu m$, $n = 25$ cells), despite significantly slower cell movement during phases of matrix stretch, rendered SSM overall faster compared to continuous migration (Fig. 3k). Thus, a higher frequency of high speed SSM events at the intermediate stiffness may in part contribute to the biphasic relationship observed between migration speed and stiffness within aligned matrices. To prevent phototoxicity during fluorescence imaging, 10-min intervals were utilized in these measurements; however, matrix recoil events captured under transmitted light (one frame per second) yielded a more accurate measurement of recoil speed

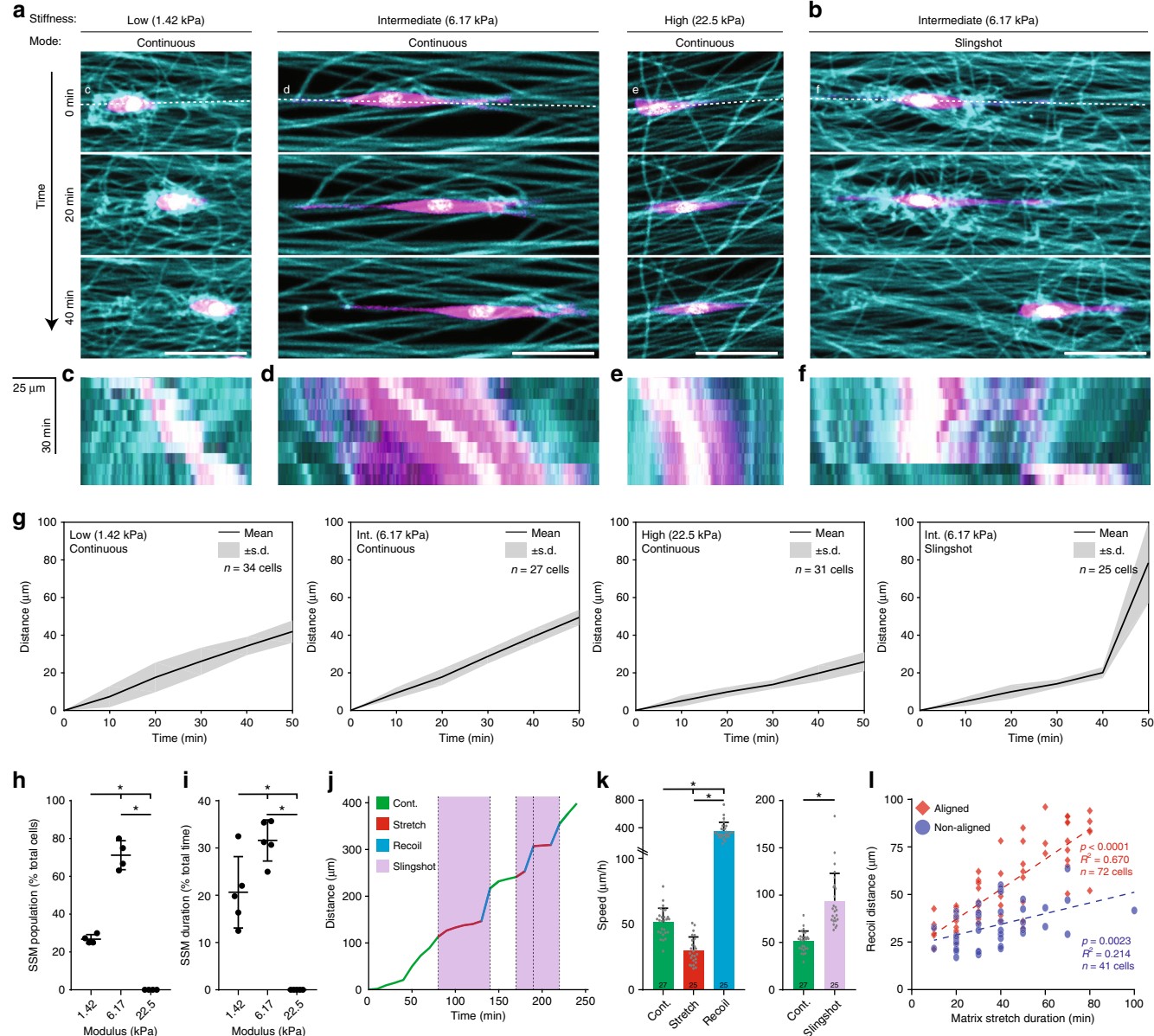

**Fig. 3** Aligned deformable matrices undergo marked deformation and promote rapid migratory events. **a**, **b** Representative time-lapse images of NIH3T3s within matrices of low, intermediate, and high stiffness utilizing continuous or slingshot migration (SSM) modes (matrix fibers (cyan), cytoplasm (magenta), and nuclei (white); scale bars: 50 μm). **c–f** Corresponding kymographs taken along the dotted lines indicated in **a** and **b**. **g** Migration distance over time for cells undergoing continuous migration and SSM. $n$ = number of cells per group as indicated within each plot; shaded region indicates standard deviation. **h** Percentage of cells observed to employ SSM out of all cells tracked over a 6 h time course ($n$ = 4 fields of view; field of view = 10 cells). **i** Percentage of time cells employed SSM as a function of matrix stiffness ($n$ = 5 fields of view; field of view = 10 cells). **j** Representative migration track of an NIH3T3 on aligned matrix of intermediate stiffness, demonstrating interconversion between continuous migration and SSM modes over its tracked lifetime. **k** Effective migration speeds of parsed phases of continuous, stretch, and recoil (left) and periods of continuous migration and SSM (combined stretch and recoil phases) (right). Continuous migration speeds were calculated from periods of continuous migration in cells that undergo SSM over its tracked duration. $n$ = number of cells per group as indicated within each bar. **l** Recoil distance (net translocation of cell) as a function of duration spent stretching the matrix ($n$ = number of cells per group as indicated in the plot). Dashed lines indicate linear correlations with indicated $R^2$ and $p$-values. All data presented as mean ± s.d.; * indicates a statistically significant comparison with $p < 0.05$ (**h**, **i**, **k**: one-way analysis of variance, l: linear regression)

of $45.2 \pm 15.1 \, \mu m \, s^{-1}$ ($n = 25$ cells) (Supplementary Fig. 3a, Supplementary Movie 3). Furthermore, cells that possessed a distinct leading edge resulted in recoil events preferentially in the direction of continuous migration prior to matrix stretch while cells with bidirectional extensions resulted in recoil events in either direction (Supplementary Fig. 3b). SSM was also observed in deformable, non-aligned matrices, and in both aligned and non-aligned matrices recoil distance positively correlated with the

duration the cell spent generating matrix stretch (Fig. 3l). For a given duration of matrix stretch, recoil distances were larger within aligned matrices, which may be due to uniaxial material stretch in aligned matrices as compared to more equiaxial stretch in non-aligned matrices.

**SSM involves coordinated matrix deformations.** Previous studies tracking fiducial markers embedded within 3D collagen

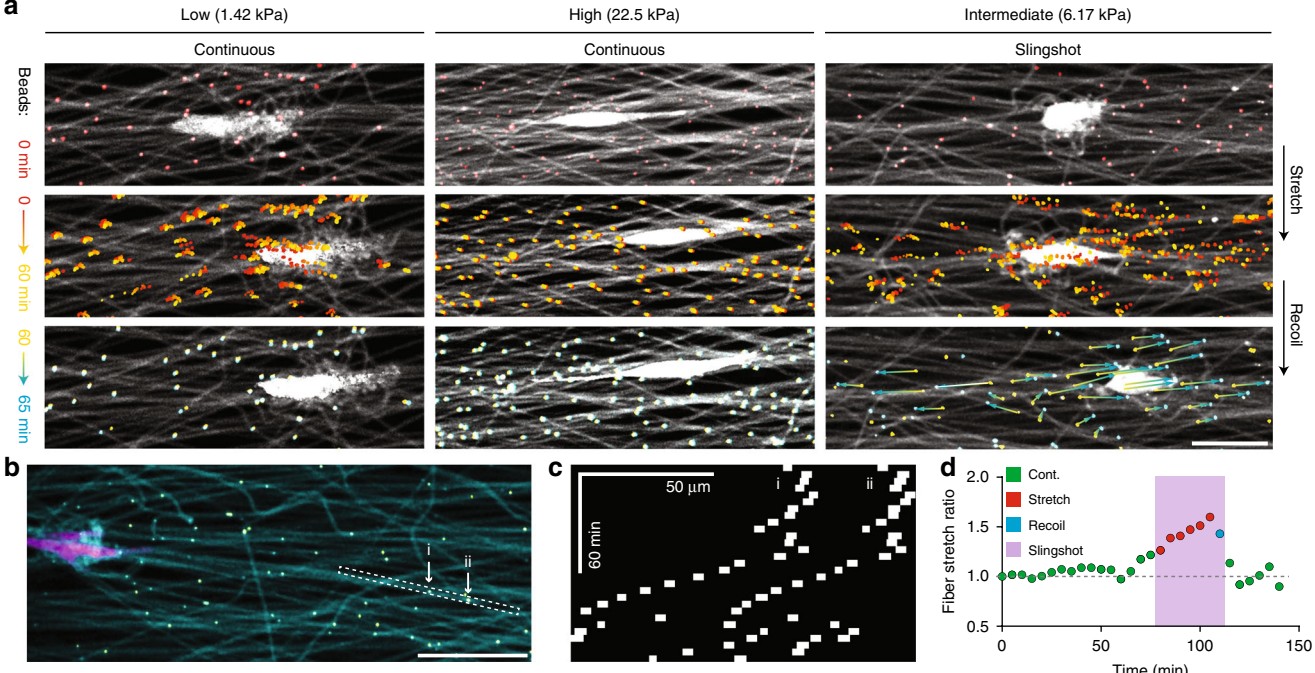

**Fig. 4** Slingshot migration involves stretch and recoil of matrix fibers. **a** Representative time-lapse images of fibers containing fluorescent microspheres, used as fiducial markers to examine matrix deformations underlying continuous and slingshot migration (scale bar: 50 μm). **b** Composite confocal fluorescence image of an NIH3T3 within an intermediate stiffness matrix (matrix fibers (cyan), cytoplasm (magenta), and fiber-embedded beads (yellow); scale bar: 50 μm). **c** Kymograph of a pair of microspheres embedded within the same fiber, as indicated in **b** used to determine fiber stretch ratio (**d**) (relative to initial distance between beads) as a function of time

hydrogels suggested cancer cells with greater metastatic potential apply uniaxial traction forces and store anisotropic strain energy within matrix deformations[17]. Adopting a similar approach, we next more closely examined matrix deformations during SSM and continuous migration by embedding fluorescent microspheres (beads) within matrix fibers. As expected, cell forces underlying continuous migration within non-deformable, high stiffness matrices resulted in negligible bead displacements (Fig. 4a, Supplementary Movie 4). In contrast, within low stiffness matrices, displacements of beads nearby continuously migrating cells increased incrementally throughout the migration track (Fig. 4a, Supplementary Movie 4). Cells undergoing SSM within intermediate stiffness matrices similarly induced incremental bead displacements during matrix stretch. However, upon matrix recoil and simultaneous translocation of the cell body, we noted large forward and rearward bead displacements emanating from the former position of the cell's trailing edge (Fig. 4a, Supplementary Movie 4).

In previous work with mesenchymal stem cells, cell traction forces applied to deformable, non-aligned matrices induced isotropic fiber recruitment (an increase in fiber density local to the cell), which led to increased focal adhesion maturation, cell spreading, and proliferation[25]. Within matrices of aligned fibers, cell morphology adopts a uniaxial phenotype likely leading to directional force generation and anisotropic fiber recruitment. However, given the implication of elastic strain energy storage and release during SSM, we also asked whether individual fibrils undergo stretch due to cell-generated traction forces. Tracking pairs of adjacent beads embedded within the same fiber revealed fibers engaged by cell adhesions undergo stretch, and release of this strain energy stored in the form of matrix stretch correlates temporally with forward motion of the cell (Fig. 4b–d). Interestingly, we noted that fiber engagement was spatially heterogeneous, where nearby non-engaged fibers did not experience

stretch (Supplementary Fig. 4a–c). Several studies in vivo and in vitro have identified mechanoreciprocity (the reciprocal mechanical interplay between cells and matrix) arising from cell contractility-generated matrix fibril stretch[1,45]. For example, cell traction forces applied to fibronectin-rich matrices induce fibronectin fibril stretch to reveal cryptic binding sites, promote growth factor bioavailability, and in turn facilitate cell adhesion, proliferation, and migration[46–48]. In our synthetic material system where such biochemical changes are absent, we show that mechanical stretch of matrix fibrils from cell traction forces can directly facilitate cell movement.

To more closely examine how matrix recoil events are initiated within SSM, we generated NIH3T3s stably expressing lifeact-green fluorescent protein (3T3-LA-GFP) or paxillin-enhanced GFP (EGFP) fusion protein and tracked their migration in aligned matrices of intermediate stiffness. Simultaneous with recoil of the matrix, we observed rupture of the cell's trailing edge resulting in residual actin- and paxillin-rich puncta tethered to the matrix rearward to the direction of cell movement (Supplementary Fig. 5a, b, Supplementary Movie 5, Supplementary Movie 6). To confirm that these paxillin-rich plaques were indeed focal adhesions, we fixed and immunostained substrates directly on the microscope stage immediately following live imaging of 3T3-LA-GFP cell migration, demonstrating localization of vinculin within the actin-rich remnants resulting from matrix recoil and tail rupture (Supplementary Fig. 6a, b). Furthermore, we investigated whether SSM occurred in 3D type I rat tail collagen hydrogels, widely utilized to model fibrous stromal tissue. Tracking 3T3-LA-GFP cells embedded within fluorescently labeled 1.0 mg mL$^{-1}$ collagen hydrogels revealed a phenotypically similar migration mode to SSM. Cells remained stagnant while deforming the surrounding matrix, followed by rapid forward migration corresponding with recoil of the matrix and lifeact-GFP puncta residing in the matrix rearward to

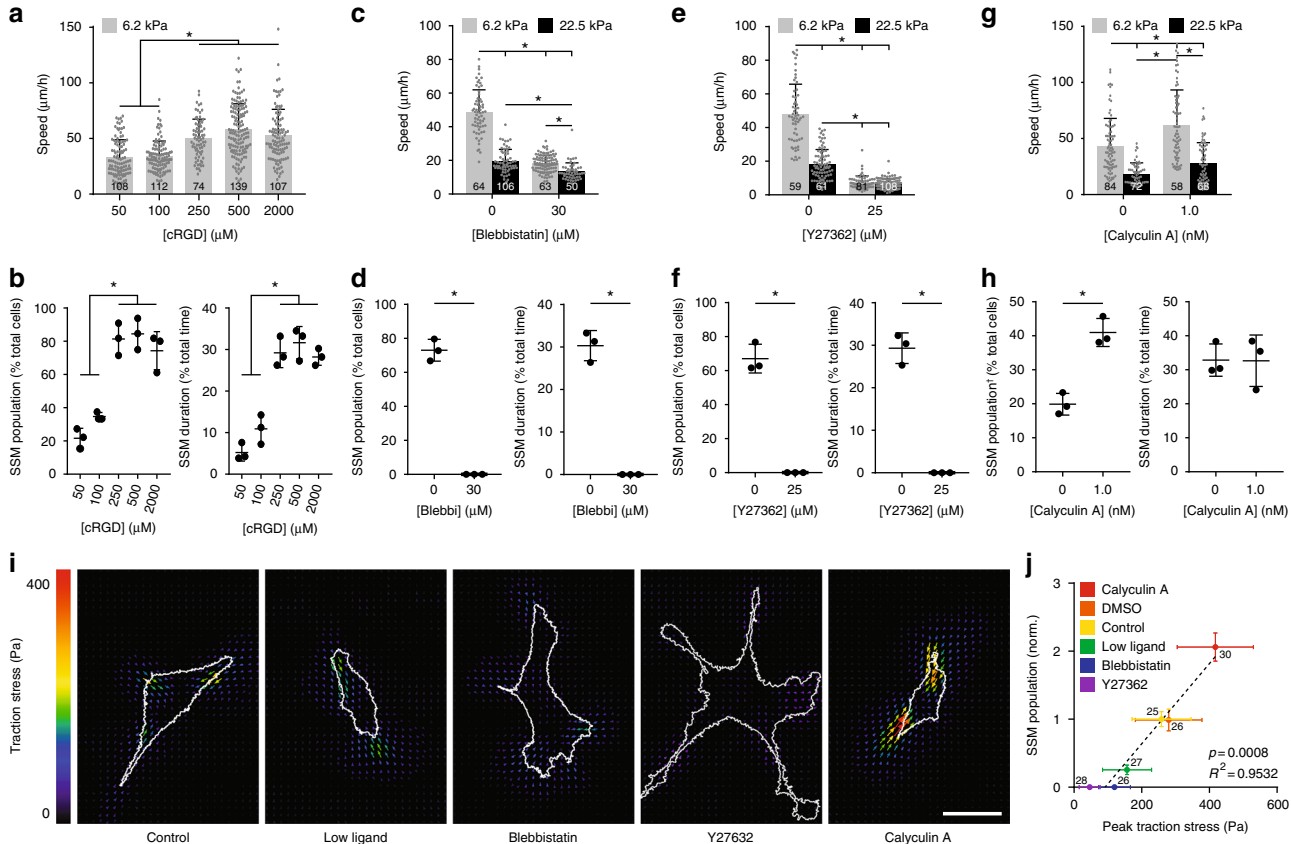

**Fig. 5** Slingshot migration (SSM) requires matrix adhesion and actomyosin contractility. **a**, **b** Migration speed, SSM population (% of cell population undergoing one or more SSM event), and SSM duration (% of tracked time of entire population spent in some phase of SSM) in aligned, intermediate stiffness (6.2 kPa) matrices with a range of cRGD functionalization. $n$ = number of cells per group as indicated within each bar (**a**); $n$ = 3 fields of view with each field of view containing 10 cells (**b**). **c**, **e**, **g** Migration speed in aligned, intermediate (6.2 kPa) and high (22.5 kPa) stiffness matrices treated with blebbistatin, Y27632, or calyculin A ($n$ = number of cells per group as indicated within each bar). **d**, **f**, **h** SSM population and slingshot duration in aligned, intermediate stiffness (6.2 kPa) matrices treated with blebbistatin, Y27632, or calyculin A ($n$ = 3 fields of view with each field of view containing 10 cells). Migration speed, SSM population, and SSM duration were quantified over a 6 h time course for cRGD, blebbistatin, and Y27 studies, and quantified over a 1 h time course for calyculin A studies (indicated with $\dagger$). **i** Representative traction stress maps of NIH3T3s on 7.9 kPa polyacrylamide hydrogels subjected to equivalent perturbations as in **a–h** (scale bar: 50 μm). **j** Slingshot population (normalized to control) as a function of peak traction stress determined by traction force microscopy. Dashed line indicates linear correlation with indicated $R^2$ and $p$-value. $n$ = number of cells per group as indicated beside each data point. All data presented as mean ± s.d.; * indicates a statistically significant comparison with $p < 0.05$ (**a–c**, **e**, **g**: one-way analysis of variance; **d**, **f**, **h**: two-tailed Student's $t$-test; **j**: linear regression)

the direction of the migration (Supplementary Fig. 7a, b, Supplementary Movie 7). These data suggest matrix recoil events are initiated by a mechanical failure of the actomyosin cytoskeleton at the cell's trailing edge. Integrins and other transmembrane proteins have been observed to remain tethered to the ECM upon ripping from the plasma membrane or rear-release during migration in vivo and in vitro[49–52]. These abrupt mechanical failure events bare resemblance to retraction-induced migration described in chick fibroblasts cultured on rigid glass substrates by Chen several decades ago[50], where tension in the cytoskeleton was hypothesized to cause elastic recoil of the cell body upon trailing edge detachment. In contrast, however, our studies employing deformable elastic substrates suggest tension stored additionally in matrix fibrils can induce substrate recoil and simultaneous cell translocation upon rupture at the trailing edge.

Given that contractility-generated traction forces induce matrix deformations and fibril stretch, we anticipated SSM would be adhesion- and contractility-dependent. Motivated by previous demonstrations that low ECM ligand density decreases cell contractility due to impaired integrin clustering and focal adhesion maturation[53,54], we first modulated the concentration of cRGD coupled to the matrix (all previous experiments utilized 500 μM cRGD). At 50 and 100 μM cRGD, SSM events decreased along with overall migration speeds (Fig. 5a, b). Treating cells with 30 μM blebbistatin (myosin II inhibitor) or 25 μM Y27362 (ROCK inhibitor) to lower actomyosin-generated contractility completely abrogated matrix deformations and SSM events, and cells were observed to migrate continuously at decreased speeds comparable to cells within non-deformable stiff matrices employing continuous migration (Fig. 5c–f, Supplementary Movie 8). Continuous cell migration requires cell contractility[44], and a decrease in continuous migration speeds on high stiffness matrices was also noted. However, the relative decrease in migration speed on intermediate stiffness matrices was significantly larger, suggesting contractility is critical for SSM due to the requirement for matrix stretch and recoil (Supplementary Fig. 8a). While 30 μM blebbistatin completely abrogated SSM events, treating cells with 5 μM blebbistatin to partially inhibit cell contractility resulted in decreased migration speed, SSM population and duration, and decreased recoil distances for a given matrix stretch duration (Supplementary Fig. 8b–d,

Supplementary Fig. 9). Finally, cells were treated with calyculin A, a relatively specific inhibitor of myosin II phosphatase, to increase overall myosin II activity[53]. Calyculin A-mediated increase of actomyosin activity led to increases in the rate of matrix stretch and SSM population and duration compared to control, and corresponded to an overall increase in migration speeds (Fig. 5g, h, Supplementary Fig. 9). In sum, these data suggest adjusting levels of intracellular force generation by modulating actomyosin contractility regulates the magnitude and frequency of rapid recoil events and resulting migration speeds.

Given the spatial heterogeneity of engaged vs. non-engaged fibers and thus non-continuum-like deformation of the matrix (Supplementary Fig. 4a–c), accurate calculation of traction forces utilizing standard TFM analysis is not currently possible in this setting. Thus, we employed 2D polyacrylamide hydrogels and the TFM method[55] to measure traction forces under the above perturbations to actomyosin contractility. Plotting SSM events as a function of peak traction stress for each condition, we found that population of cells that utilized SSM significantly correlates with contractility-generated traction forces (Pearson correlation: $R^2 = 0.9532$; $p$-value = 0.0008; linear regression) (Fig. 5i, j, Supplementary Fig. 11a). Taken together, these studies suggest SSM involves a dynamic force balance between actomyosin contractility and tension generated from matrix stretch. When matrix tension rises beyond the critical force level that trailing edge adhesions can bear, failure at the interface between focal adhesions and the cytoskeleton triggers a sudden imbalance of forces and rapid forward translation of the cell with matrix recoil.

**Matrix stiffness for optimal SSM scales with traction forces.** To investigate whether other cell types employ SSM, we performed similar migration studies with a variety of metastatic cancer cells, adult mesenchymal stem cells, and fibroblasts. An initial screen utilizing aligned matrices at intermediate stiffness revealed cell types that utilize adhesion-dependent mesenchymal-like migration indeed also employ SSM. However, the degree of matrix deformations generated at this stiffness, which led to maximal SSM duration for NIH3T3s, varied considerably between these cell types (Supplementary Fig. 10). Since previous work has shown optimal migration speeds between cell types shift with substrate stiffness[56], we hypothesized these variations may be due to intrinsic differences in contractility-generated traction forces between cell types. Thus, we screened a range of stiffnesses for each cell type and found that the optimal SSM stiffness for metastatic breast (MDA-MB-231), squamous cell carcinoma (UM-SCC-74B), fibrosarcoma (HT1080), C2C12 myoblast, human mesenchymal stem cell, and human foreskin fibroblast significantly positively correlated with each cell type's peak traction stress (Pearson correlation $R^2 = 0.7774$; $p$-value = 0.0087; linear regression) (Fig. 6a, b, Supplementary Fig. 11b). These studies suggest that the matrix stiffness that optimizes matrix stretch and subsequent recoil (and thus the frequency of SSM events) scales directly with contractility-generated traction forces.

## Discussion

This work characterizes a previously undescribed cell migration mode whereby cell contractility-mediated reorganization and stretching of matrix fibrils directly contributes to rapid cellular movement. In contrast to continuous migration, involving short iterative cycles of extension and adhesion of the leading edge followed by contraction of the cell body and detachment of the cell rear, SSM consists of two distinct phases: (1) matrix stretch and (2) matrix recoil. During matrix stretch, extension of cell protrusions and actomyosin contractility actively recruit and stretch nearby fibers to store elastic strain energy within the matrix. Once tension in the matrix surpasses maximal forces cell-ECM adhesions can withstand (potentially due to force fluctuations at adhesions[57]), failure at trailing edge adhesions initiates sudden matrix recoil, the release of matrix stored strain energy, and concomitant cell translocation (Fig. 6c). This migration mode proves highly mechanosensitive and requires simultaneous production of both matrix stretch and tension: softer matrices undergo pronounced stretch but fail to generate sufficient tension to induce recoil; conversely, stiff matrices may experience high forces, but yield insignificant stretch (Fig. 6e). We find that a variety of mesenchymal cells adopt this migration mode, and furthermore, that this event occurs optimally at a matrix stiffness that scales with the cell type's baseline contractility.

Though well accepted that cell migration involves a coordination of molecular events that generates ECM tractions and contractile forces that move the cell body forward, this work extends the role of these processes to actively deforming the matrix to elicit cell translocation. The growth and decay of focal adhesions and resulting dynamics of force transmission to the ECM have previously been described using motor-clutch models, which yield a qualitatively similar biphasic response of migration speed to matrix stiffness that is likely at play in our studies[21,39,40,42,43]. Such models recently have been extended to fibrous matrices[58], and further adopting these models to incorporate matrix heterogeneity and cell translocation due to large-scale matrix stretch may provide additional insights into the observations described here. Besides the actomyosin cytoskeleton and focal adhesions, other cellular components that are mechanically contiguous along the matrix-cell-matrix unit (Fig. 6d) may mediate force transmission and the dynamics underlying this phenomenon. For example, the nucleus is mechanically coupled to the ECM via linker of nucleoskeleton and cytoskeleton complexes, actomyosin stress fibers, and their termination at focal adhesions[59] (Fig. 6d). Although not explored here, perturbations to the nucleoskeleton or focal adhesion complexes may provide additional points of control over the force dynamics involved in SSM. Advances in single-cell isolation and RNA-sequencing techniques[60] could help identify the underlying signaling pathways responding to local ECM changes as cells actively deform the surrounding ECM during cell migration.

Cells encounter heterogeneity in ECM mechanical properties and architecture as they migrate through fibrillar interstitial tissues in vivo. How cells interpret sundry biophysical cues and adopt migration strategies to efficiently navigate through the ECM remains an ongoing effort. Recent advances in intravital imaging techniques have provided tremendous insight into visualizing cell migration in native tissue settings[4]. However, challenges with cell and ECM labeling, image capture rates, and imaging depth may limit high spatiotemporal resolution visualization of highly dynamic cell-ECM interactions such as those described here. It has yet to be determined whether matrix deformations directly contribute to rapid motion of cells in vivo, although recent intravital studies describe buckling of collagen fibers by fibroblasts within tumors[61]. Certainly, cell migration concurrent with deformation of fibrous ECM occurs developmentally as mesodermal cells migrate along pliable fibronectin fibrils from the blastopore to the animal pole[62], in loose connective tissues, such as the stroma surrounding mammary glands during tumor cell escape[23], and upon early stages of wound healing as fibroblasts colonize provisional fibrin-rich ECM and exert contractile forces to close the wound gap[63]. During morphogenesis of the forming heart tube, large-scale tissue

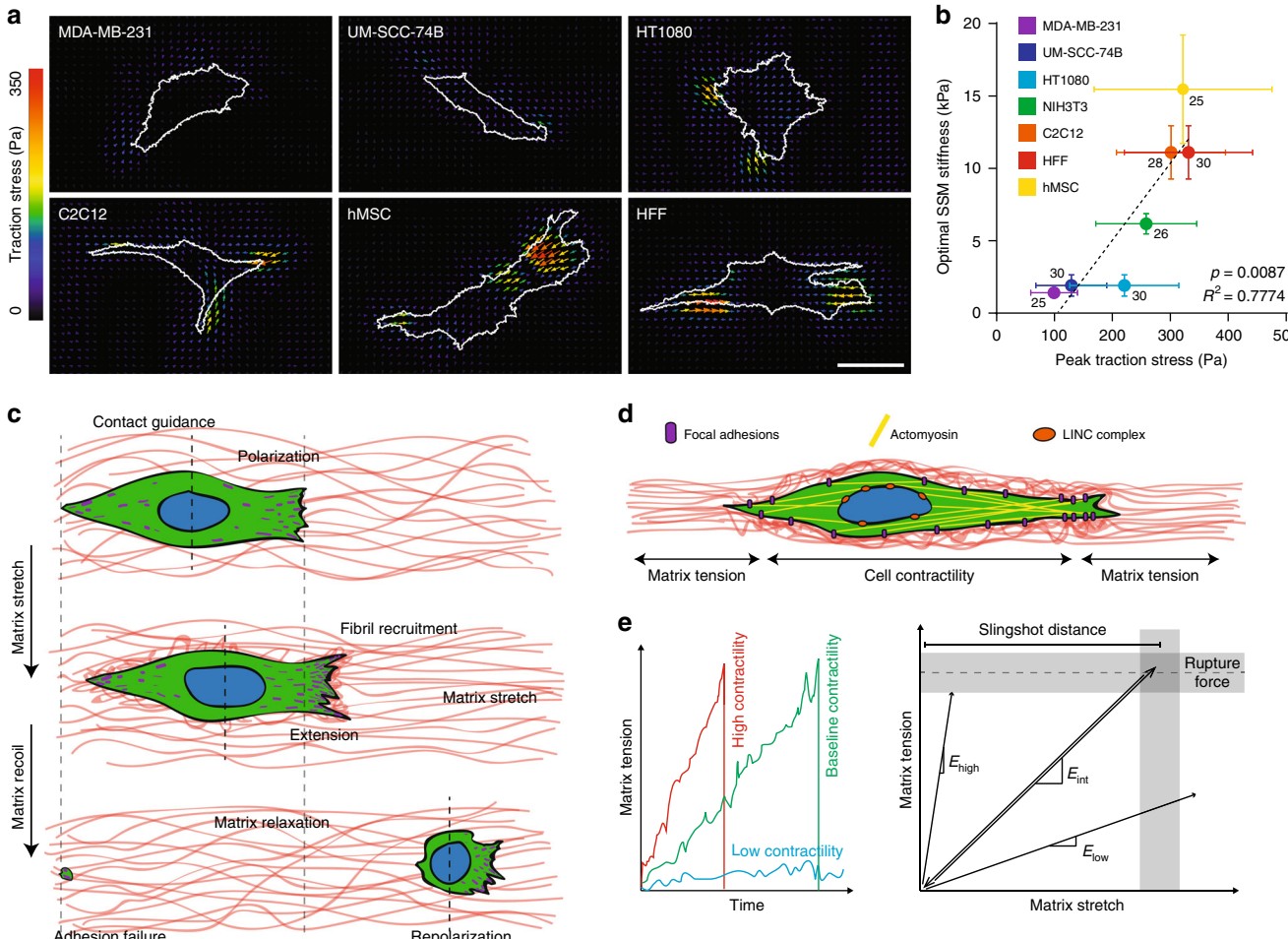

**Fig. 6** Contractile cell types employ slingshot migration in a matrix stiffness-dependent manner. **a** Representative traction stress maps of MDA-MB-231 (breast adenocarcinoma), UM-SCC-74B (head and neck squamous cell carcinoma), HT1080 (connective tissue fibrosarcoma), C2C12 (mouse myoblast cell line), hMSC (primary adult mesenchymal stem cells), and HFF (primary human foreskin fibroblasts) on 7.9 kPa polyacrylamide hydrogels (scale bar: 50 μm). **b** Matrix stiffness leading to maximal slingshot duration for each cell type as a function of peak traction stress for each cell type, as determined by traction force microscopy. $n$ = number of cells per group as indicated beside each data point. Data presented as mean ± s.d. Dashed line indicates linear correlation with indicated $R^2$ and $p$-value (linear regression). **c** Top-down schematic depicting key steps underlying slingshot migration in aligned fibrous matrices, including contact guidance-mediated polarization (top), protrusion extension simultaneous with fibril recruitment and matrix stretch (middle), and adhesion failure inducing rapid recoil and cell translocation (bottom). Theorized adhesion distributions are shown in purple. **d** Schematic highlighting force balance between cell-generated contractile forces and tension in the matrix. **e** Proposed time evolution of matrix tension as a function of cell contractility (left) and relationship between matrix tension and stretch at different levels of matrix stiffness (right). Note: these plots are theorized, not experimental

movements convect endocardial progenitors to their target destination[64,65]. In similar fashion, recent work patterning mesenchymal condensation demonstrated programmable movement of non-contractile epithelial and endothelial cells via fibroblastic compaction of the ECM[66]. In contrast to these observations, in our studies the same cells inducing local matrix deformations were translocated by subsequent matrix motion. Furthermore, considering heterotypic cell interactions, and given recent observations of mechanical forces underlying heterotypic cell interactions mediating tumor cell escape[11], it is possible that more contractile cell types could induce rapid migration of another cell type via deformations to the matrix. Future work examining mechanoreciprocal relationships between multiple cell types as well as the ECM using such biomimetic in vitro models will grow our understanding of cell migration and help inform future intravital imaging efforts.

In these studies, we modulated stiffness at the single-fiber and bulk matrix scales by tuning photocrosslinking of the

polymer network within each matrix fiber. While bulk mechanical testing demonstrated orthogonal control over fiber alignment and bulk stiffness of these matrices, cells likely sense and respond to local matrix mechanics and ligand availability[25,67], and in particular, anisotropic mechanical behavior intrinsic to matrix fibers additionally influenced by fiber alignment at the population level. We performed TFM on 2D PAAm hydrogels to determine baseline contractility, but it is likely that cell-generated forces are highly dependent upon matrix topography and mechanics. Continued development of technologies such as magnetic bead rheology, TFM, and computational modeling will aid in our understanding of how cells reciprocally interpret the architecture and mechanics of fibrous matrices, exert traction forces, and in turn how these forces yield stresses within the ECM to engender reorganization and deformation. Such integrated approaches along with tunable biomimetic ECM models could aid in understanding and coordinating developmental migratory processes, designing biomaterials that rapidly recruit repair cells to

wound sites, or identifying targeted therapeutics to arrest primary cancer cell egress through the surrounding tumor stroma.

## Methods

**Reagents**. All reagents were purchased from Sigma-Aldrich and used as received, unless otherwise stated.

**Synthesis of DexMA**. Dextran (molecular weight 86,000 Da, MP Biomedicals, Santa Ana, CA) was modified with methacrylate groups as in van Dijk-Wolthuis et al.[68]. In brief, dextran (30 g) and 4-dimethylaminopyridine (3 g) were dissolved in anhydrous dimethyl sulfoxide (DMSO; 150 mL) overnight. Next, glycidyl methacrylate (36.9 mL) was added under vigorous stirring, heated to 45 °C and allowed to react for 24 h. The solution was then cooled to 4 °C and precipitated into chilled (4 °C) 2-proponal (1 L). The crude product was recovered by centrifugation at a speed of $3000 \times g$, re-dissolved in milli-Q water and dialyzed against milli-Q water for 3 days with two solvent exchanges daily. The solution was then lyophilized for 3 days to obtain the pure product, which was characterized by [1]H-nuclear magnetic resonance spectroscopy in $D_2O$. The degree of methacrylate functionalization was calculated as the ratio of the proton integral (6.174 and 5.713 ppm) and the anomeric proton of the glycopyranosyl ring (5.166 and 4.923 ppm). As the signal of the anomeric proton of α-1,3 linkages (5.166 ppm) partially overlaps with other protons, a pre-determined ratio of 4% α-1,3 linkages was assumed[68] and the total anomeric proton integral was calculated solely based on the integral at 4.923 ppm. A methacrylate per dextran repeat unit ratio of 0.7 was determined.

**DexMA fiber network fabrication**. 3D networks of suspended DexMA fibers were generated as in Baker et al.[25]. In brief, DexMA was dissolved at 0.475 g mL$^{-1}$ in a 1:1 mixture of milli-Q water and dimethylformamide with 0.005% Irgacure 2959 photoinitiator. This polymer solution was utilized for electrospinning within an environment-controlled glovebox held at 21 °C and 30–40% relative humidity. Electrospinning was performed at a flow rate of 0.5 mL h$^{-1}$, gap distance of 7 cm, and voltage of $-4.3$ kV, onto a collecting surface of oppositely charged (4.9 kV) parallel electrodes with varying separation distance to control alignment (Fig. 1b, c). Methacrylated rhodamine (0.5 mM; Polysciences, Inc., Warrington, PA) was incorporated into the electrospinning solution to fluorescently visualize DexMA fibers and fluorescent microspheres (stock diluted 1:20 in polymer solution; ThermoFisher Scientific, Waltham, MA) were added for matrix deformation studies. Fibers were collected on microfabricated PDMS (Sylgard 184, Dow-Corning, Midland, MI) arrays of wells to isolate cell-perceived mechanics of suspended fiber networks. In brief, silicon wafer masters possessing SU8 photoresist (Microchem, Westborough, MA) were produced by standard photolithography and used to generate PDMS stamps. Following silanization with trichloro (1H,1H,2H,2H-perfluorooctyl)silane, stamps were used to emboss uncured PDMS onto oxygen plasma-treated coverslips. PDMS well arrays were then surface functionalized with 3-(trimethoxysilyl)propyl methacrylate (Gelest, Inc., Morrisville, PA) via vapor deposition in a vacuum oven (60 °C and 14.7 PSI) to facilitate DexMA fiber adhesion. After electrospinning over PDMS well substrates, suspended network samples were first primary crosslinked under UV light (100 mW cm$^{-2}$) to stabilize fibers. Samples were then hydrated in a photoinitiator solution of 1 mg mL$^{-1}$ Irgacure 2959 in milli-Q water and exposed to a secondary crosslinking of varying durations of UV light (100 mW cm$^{-2}$) to control the degree of crosslinking and resulting stiffness (Fig. 1d). To promote cell adhesion, cyclo [RGDfK(C)] (cRGD, Peptides International, Louisville, KY) was coupled to methacrylates along the DexMA backbone via Michael-type addition chemistry.

**Mechanical testing**. To determine the Young's modulus of suspended fiber networks, microindentation with a rigid SU8 cylinder (1 mm diameter and 1 mm tall) was performed on a commercial Cell Scale Microsquisher (CellScale, Waterloo, Ontario). Cylinders of SU8 photoresist were microfabricated and affixed to pure tungsten filaments (0.156 mm diameter and 59.6 mm length). Fiber networks were generated over 2 mm diameter circular PDMS well substrates, crosslinked under the conditions described above, and indented to a depth of 150 μm at a strain rate of 0.05%/s to determine a Young's modulus. Young's modulus was approximated assuming an elastic membrane using the following equation:

$$F = \frac{Et\pi\delta^3(r_o^2 - r_i^2)}{2(r_o - r_i)^4(1-\nu)} \quad (1)$$

where $t$ is the membrane thickness (determined by confocal microscopy for each sample, ranging from 20 to 40 μm), $r_o$ is the membrane radius (1 mm), $r_i$ is the indenter radius (0.5 mm), and $\nu$ is the Poisson ratio (0.5), $F$ is the indentation force, $\delta$ is the indentation depth, and $E$ is Young's modulus.

To determine Young's modulus of single fibers, three-point bending tests were performed using AFM as in ref. [25]. Single fibers were electrospun on 200 μm wide by 200 μm tall microfabricated PDMS troughs. Fibers were hydrated and crosslinked to varying degrees by UV light exposure as described above, and deformed by an AFM tip (0.06 N m$^{-1}$) loaded with a 25 μm diameter bead

positioned centrally along the fiber's length. Young's modulus was calculated from the resulting load-displacement curves using known equations for a cylindrical rod undergoing three-point bending with fixed boundaries[69].

**Cell culture and biological reagents**. NIH3T3 fibroblasts (ATCC CRL-1658) were cultured in high-glucose Dulbeco's modified Eagle's medium (DMEM) containing 1% penicillin/streptomycin, L-glutamine, and 10% bovine serum (ThermoFisher Scientific, Waltham, MA). Cells were passaged upon achieving confluency at a 1:4 ratio and used for studies until passage 20. For all studies, cells were trypsinized, counted, and seeded onto substrates at a density of 1500 cells per cm$^2$. Human foreskin fibroblasts (a gift from Kenneth Yamada), human mesenchymal stem cells (Lonza PT-2501), C2C12 (ATCC CRL-1772), UM-SCC-74B (a gift from Thomas Carey), HT1080 (ATCC CCL-121), and MDA-MB-231 (ATCC HTB-26) cells were cultured in DMEM containing 1% penicillin/streptomycin, L-glutamine, and 10% fetal bovine serum (Atlanta Biologics, Flowery Branch, GA). Fluorescently labeled collagen hydrogels (Corning) were prepared as in Doyle et al.[70]. NIH3T3 fibroblasts were encapsulated at 250,000 cells per mL in a 1.0 mg mL$^{-1}$ collagen hydrogel solution prepared as in Kuntz and Saltzman[71], and pipetted into glutaraldehyde-functionalized PDMS gaskets (6 mm diameter × 2 mm height) bonded to a 35-mm MatTek dish. Cells were imaged by confocal fluorescence time-lapse microscopy >200 μm away from glass or PDMS boundaries. For vinculin immunostaining, samples were simultaneously fixed and permeabilized in 2% paraformaldehyde in microtubule-stabilizing buffer for 15 min at room temperature to extract cytoplasmic vinculin. Samples were then blocked for 1 h in 10% fetal bovine serum, incubated for 1 h with primary mouse monoclonal anti-vinculin (1:500, Sigma-Aldrich V9264), and incubated for 1 h with secondary AlexaFluor 647 goat anti-mouse IgG (H + L) (1:1000, Life Technologies A12379) sequentially at room temperature. AlexaFluor 488 phalloidin (Life Technologies) and 4′,6-diamidino-2-phenylindole (Sigma-Aldrich D8417) were utilized to visualize F-actin and nucleus, respectively.

**Microscopy and image analysis**. Time-lapse microscopy was performed on an LSM800 laser scanning confocal microscope (Zeiss). Unless otherwise specified, migration experiments were imaged 6 h after cell seeding at 10 min frame intervals over 8 h. Immediately prior to imaging, cell nuclei were labeled with Hoechst33342 (3 μg mL$^{-1}$) for 10 min and rinsed once with media for 10 min. Following raw image export, cell nuclei were tracked with a custom Matlab script predicated on the IDL Particle Tracking code[72]. Briefly, parameters to threshold and locate the centroids of cell nuclei were identified and applied uniformly across the entire data set. Nuclei centroids in serial images were linked using IDL to define migration tracks.

Migration speed was calculated as total tracked distance over total tracked duration. Deviation was calculated as the minimum distance, for each time point, between the tracked position of a cell's nucleus and a line of best fit to the cell's overall track (determined by linear regression to model a perfectly non-deviating cell) and normalized to its total tracked duration using the following equation:

$$\text{Deviation} = \frac{\sum_1^n d_n}{n} \quad (2)$$

where $n$ is the number of tracked positions and $d_n$ is the distance between a cell's tracked position and line of best fit. Cell track orientation was then calculated by the angle ($-90$ to $+90$) between the line of best fit and a reference line (direction of fiber alignment). SSM population was calculated as the percentage of all tracked cells that exhibit a slingshot event at some point during each cell's tracked migration. Values for SSM population were determined over a 6 h time course in all studies, with the one exception of the calyculin A studies (described below), which were performed over a 1 h time course. This is due to the observation that heightened contractility from dosing with 1 nM of calyculin A prevented cells from respreading following SSM events, and thus they were subsequently non-migratory for the remainder of the time-lapse. SSM duration was calculated as the percentage of time cells underwent SSM divided by the total tracked time. Recoil distances were measured from the centroid of the cell nucleus at the final frame prior to recoil, to the new position of the nuclear centroid immediately following recoil and translocation of the cell. Fluorescent microsphere tracks in deformation studies and kymographs were generated in FIJI. FibrilTool[73] was utilized to quantify the coefficient of alignment of fibers within electrospun matrices.

**Pharmacologic contractility perturbations**. Blebbistatin, calyculin A, and Y27362 (Santa Cruz Biotechnology, Dallas, TX) were diluted to working stock concentrations and stored per the manufacturer protocol. Blebbistatin was utilized at 30 or 5 μM, calyculin A at 1.0 nM, and Y27362 at 25 μM; all pharmacologics were diluted in DMSO. For blebbistatin and Y27362 experiments, samples were treated with the respective dose for 1 h and imaged for 8 h at 10 min intervals. For calyculin A experiments, samples were treated for 30 min and imaged for 1 h at 10 min intervals. Beyond time points of 2 h post treatment, calyculin A induced a non-migratory, rounded cell morphology, consistent with cells plated on tissue culture plastic. Cells that exhibited a non-migratory, rounded cell morphology from calyculin A treatment were excluded from cell migration analysis.

**Traction force microscopy**. PAAm hydrogels (Young's modulus, $E = 7.9$ kPa) were prepared as in Aratyn-Schaus et al.[55]. Fluorescent microspheres (0.2 μm diameter; Thermofisher) were mixed into the PAAm solution at 1% v/v before polymerization. After polymerization, the PAAm surface was derivatized with sulfo-SANPAH (Proteochem, Hurricane, UT) and functionalized by 2–3 h incubation with type 1 rat tail collagen (50 μg mL$^{-1}$; Corning) diluted in phosphate-buffered saline (PBS). For all traction force experiments, cells were seeded at a density of 500 cells per cm$^2$ and allowed to adhere for 12 h before TFM was performed. Fluorescent images of 10–15 cells and embedded beads were captured at ×20 magnification. Images were taken at each cell position, and again after cells were lysed with 5% v/v sodium dodecyl sulfate in PBS. Data analysis was performed using an ImageJ plugin suite created by Tseng et al.[74], which was adapted from Dembo and Wang[75]. This suite consists of stack alignment, particle image velocimetry, and Fourier transform traction cytometry (FTTC). For FTTC, the Poisson's ratio of the PAAm gel was assumed to be 0.5 and a regularization parameter of $2 \times 10^{-9}$ was used. The outputted traction force vector maps were analyzed using custom Matlab script to determine the peak traction generated by each cell. For contractility perturbation studies, blebbistatin (30 or 5 μM) and Y27362 (25 μM) were added 1 h prior to imaging, and calyculin A (1.0 nM) was added 2 h prior to imaging. For decreased ligand studies, 0.1 μg mL$^{-1}$ collagen was utilized.

**Lentivirus production**. cDNA for pLenti CMV GFP Puro (658-5) was a gift from Eric Campeau and Paul Kaufman (Addgene plasmid #17448[76]). Portions of paxillin-pEGFP (a gift from Rick Horwitz, Addgene plasmid # 15233) or mRuby-Paxillin-22 (a gift from Michael Davidson, Addgene plasmid #55877) were subcloned into a modified version of the pRRLSIN.cPPT.PGK-GFP.WPRE (a gift from Didier Trono, Addgene plasmid # 12252) by restriction enzyme cloning. pLenti.PGK.LifeAct-GFP.W was a gift from Rusty Lansford (Addgene plasmid # 51010). To generate lentivirus, plasmids were co-transfected with pCMV-VSVG (a gift from Bob Weinberg, Addgene plasmid #8454), pMDLg/pRRE, and pRSV-REV (gifts from Didier Trono, Addgene plasmid #12251 and #12253[77,78]) in 293T cells using the calcium phosphate precipitation method[79]. Viral supernatants were collected after 48 h, concentrated with PEG-it$^{TM}$ (System Biosciences, Palo Alto, CA) following the manufacturer's protocol, filtered through a 0.45 μm filter (ThermoFisher Scientific Nalgene, Waltham, MA), and stored at −80 °C. Viral titer was determined by serial dilution and infection of NIH3T3s in the presence of 10 μg mL$^{-1}$ polybrene (Santa Cruz Biotechnology, Dallas, TX). Titers yielding maximal expression without cell death or detectable impact on cell proliferation or morphology were selected for studies.

**Statistics**. Statistical significance was determined by one- or two-way analysis of variance or two-sided Student's $t$-test where appropriate, with significance indicated by $p < 0.05$. Sample size is indicated within corresponding figure legends and all data are presented as mean ± s.d.

**Reporting summary**. Further information on experimental design is available in the Nature Research Reporting Summary linked to this article.

## Code availability
Custom MATLAB code (written with version R2014b) utilized to track cell nuclei is available at DOI 10.6084/m9.figshare.7591205.

## Data availability
The data that support the findings of this study are available from the corresponding author upon reasonable request.

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

## Acknowledgements

We thank Christopher Chen, Matthew Kutys, Edna Cukierman, and Farzan Beroz for helpful discussions. This work was supported in part by the National Institutes of Health (HL124322). W.Y.W. acknowledges financial support from the University of Michigan Rackham Merit Fellowship and the National Science Foundation Graduate Research Fellowship Program (DGE1256260). C.D.D. acknowledges financial support from the National Science Foundation Graduate Research Fellowship Program (DGE1256260). B.M.B. acknowledges financial support from a Ruth L. Kirschstein National Research Service Award (EB014691).

## Author contributions

W.Y.W. and B.M.B. designed the experiments. W.Y.W and D.L. conducted experiments and analyzed the data. C.D.D. performed traction force microscopy experiments. W.Y.W. and B.M.B. wrote the manuscript. All authors reviewed the manuscript.

## Additional information

**Competing interests:** The authors declare no competing interests.

