## [Peer Review File · Nature Communications]

Reviewers' comments:

Reviewer #1 (Remarks to the Author):

The manuscript by Wang et al. describes a purported new mode of migration that is termed "slingshot." To reach this conclusion they utilize advanced fabrication methods to create fibrous environments where the mechanical rigidity, alignment, and biomolecular surface coating identity/concentration can be independently controlled. This is a very interesting experimental system, and provides a potentially rich platform for fundamental investigations into cellular processes that depend on these environmental parameters, such as cell migration. The most interesting results, in my view, are the possible optimal stiffness (Fig. 2E), the "slingshot" behavior, and its dependence on actomyosin contractility-adhesion. I will not that the "slingshot" behavior described here seems like the actomyosin-dependent "load-and-fail" dynamics described by Chan and Odde (Science, 2008) for soft environments, although the present study goes further to relate these dynamics specifically to cell migration. The experiments are all well-executed and presented in a clear manner. Where I thought the manuscript was lacking was in its theoretical underpinnings, as the theorizing at the end of the manuscript did not have a mechanistic model with which to contextualize the findings. Given that recent studies have increasingly integrated such mathematical-computational modeling, the present manuscript was therefore not as sophisticated analyses (see reference below). Overall, this is an interesting study with important new observations on cell migration dynamics in state-of-the-art controlled environments, although there is room to improve by integration of mathematical-computational modeling.

Comments:

1) Lack of modeling makes the study not so cutting edge. For example, these recent studies from a number of different groups have all integrated mathematical-computational modeling that served not just for illustration purposes, but instead helped design experiments and interpret otherwise puzzling phenomena.

Stroka et al., Cell, 2014
Elosegui-Artola et al., Nat Materials, 2014
Chaudhuri et al., Nat Comm, 2015
Sunyer et al., Science, 2016
Elosegui-Artola et al., Nat Cell Biol, 2016
Elosegui-Artola et al., Nat Cell Biol, 2016
Bangasser et al., Nat Comm, 2017
Klank et al., Cell Reports, 2017
Oria et al., Nature, 2017
Weinberg et al., Biophys J, 2017
Mekhdjian et al., Mol Bio Cell, 2017
Estabridis et al., Ann Biomed Eng, 2018
Gong et al., PNAS, 2018

2) Fig. 3E. Legend, part (g) is labeled (f)

3) Fig. 5A. Why isn't the control (0 μ M) shown?

4) Fig. 6B. The correlation with peak traction force and optimal stiffness is in Bangasser et al., Biophys J, 2013, Fig. 4B.

5) Need journal name: Drifka, C. R. et al. Periductal stromal collagen topology of pancreatic ductal adenocarcinoma differs from that of normal and chronic pancreatitis. 28, 1470–1480 (2015).

6) Aratyn-Schaus, Y., Oakes, P. W., Stricker, J., Winter, S. P. & Gardel, M. L. Preparation of Complaint Matrices for Quantifying Cellular Contraction. J. Vis. Exp. 1–6 (2010).

doi:10.3791/2173. "Compliant" is misspelled

Reviewer #2 (Remarks to the Author):

This paper should be of interest to the readers, for both its methods and results. Synthetic electrospun fibers were suspended across silicone wells, to provide a 3D fibrous environment that mimics the extracellular matrix. The approach avoids mechanical input from an underlying surface, thereby simplifying the interpretation. Methods were also developed to control the stiffness, orientation, and adhesiveness of the fiber. Cells embedded in these fibers showed an interesting mode of migration, referred to as sling shot migration, where surges of movement took place upon the release of tail, coupled to the recoil of fibers.

The study is not without significant weaknesses. First, the physiological significance of slingshot migration is questionable, as no such movement has been found in intravital imaging. The process requires a specific range of fiber density, configuration, orientation, and adhesiveness, which may be difficult to find in vivo. In addition, the percentage of cells undergoing slingshot migration appeared very low, around 1% (the authors should be more upfront about this; the text in line 132 simply stated "a subset of cells"). The process also involved highly compressible properties of dextran methacrylate fibers that are not found in native ECM fibers.

Second, slingshot migration may be very similar if not identical to "retraction induced protrusion" as described decades ago by Chen (ref. 61), since pronounced tail retraction appeared to accompany the process. Evidence was weak that slingshot migration was driven by the recoil of elastic fibers, as proposed on line 192-193, rather than the retraction of tail as for retraction induced protrusion. Chen's paper, cited amongst other references, needs to be addressed carefully. Major differences need to be presented in order to justify the present finding as a separate phenomenon.

Specific Points:

1. Stiffness was measured with a bulk compression approach, which is likely very different from how a cell probes local fibers. Therefore the relevance of Young's modulus as presented is questionable.
2. The presence of focal adhesions on synthetic fibers is not convincing (Supplemental Figure 4), which in turn affects the statements involving focal adhesions.
3. Also, the suggestion that "the recoil event is initiated by mechanical failure of trailing edge focal adhesions", based on the appearance of residual focal adhesions and timescale, is not convincing.
4. Clarify the duration for measuring the frequency of slingshot migration. It makes a major difference whether the 1% frequency was found over 10 minutes or 24 hours.
5. The effect of Calyculin on migration speed looks borderline (Figure 5d; $p \sim 0.05?$). The two-fold increase of slingshot frequency is more convincing, which then raises the question of why the average speed showed only a marginal difference.
6. The comparison between fiber-mediated contact guidance and stiffness-induced polarization (lines 114-115) is misleading, since the two involve different spatial and temporal frames. The former affects mainly migration persistence while the latter affects mainly the steering of protrusions.

Reviewer #3 (Remarks to the Author):

In this work, the authors electrospun DexMA fibers to generate well-controlled synthetic fibrous ECM and observed cell migration. On stiff matrices, cell migration is lower and on intermediate and soft matrices, cell migration is higher. Furthermore, the authors report a distinct mode of motility

on softer matrices where the cells spread, contract the matrix, and then when a tension threshold is reached the rear adhesion will break and cell will "slingshot" forward using the front adhesions and recoil of the fibers. The authors further characterize this mode of migration and use various inhibitors and additional cell lines to reveal a more general correlation between traction forces and slingshot migration (SSM) frequency. I could not recommend the publication of the manuscript in Nature Communications at this stage. I have many major points that should be addressed.

Major points:

I found several claimed findings to be rather overstated. For instance, the so-called biphasic behavior of cell speed as a function of the stiffness should be clarified. Additional points between 0 and 7 kPa would be required to claim that there is indeed a biphasic behavior. I would be very cautious in the interpretation of the graph. Another example is the discussion about FAs based on Sup Fig 4. The FA staining is far from being convincing. It remains difficult to advance that there is indeed a liberation of the entire FAs. The paxillin staining is localized everywhere within the entire cell.

The porosity of the fibrillar matrix displays heterogeneities. How does it affect the measurements? Do the authors compare migration events over similar porosities for the different stiffness?

Slingshot frequencies should be expressed in units as a function of time, for example #events per time (otherwise I do not know common these events are; the authors observe ~70% of cells over 8 hours imaging for intermediate stiffness in Fig 3D but that's the only information given).

The relaxation mechanism does not appear very clear to me. On intermediate stiffness (fig3), it seems that the fibers are bundled and stretched at the back of the cell after its movement. Could the authors clarify this point? How do they measure the relaxation and the matrix stretch? This point should be clarified as well.

SSM seems to have very little effect on the overall migration speed of cells in intermediate stiffness matrices (Fig 2B vs 3F). Also, Fig. 3F is missing a bar.

Treatment with Calyculin A doubles the SSM frequency and increases the distance, but the overall speed of migration has a very modest change (Fig. 5D). Taken together with the previous 2 comments, it seems this mode of migration has a small contribution towards the overall migration speed and does not explain why cells are migrating faster on soft and intermediate matrices than on stiff.

One simple explanation for cells moving much slower in stiff 3D fibers is that the cell is unable to deform the fibers to make space for the cell to squeeze through, whereas compliant fibers would allow for this. Have the authors checked whether the cells are on top of the fiber matrix (2D) or if the cells are fully embedded within the fibers (3D)? (Fig S1C has a confocal image and it appears the cell is fully embedded, but it's not clear how general this is).

The authors observe stretching of individual fibers by tracking bead positions as fiducial markers. In a 3D collagen system, the Young's modulus of a fibril can be ~GPa thus it is unlikely this would be the case (would this affect SSM?). What is the Young's modulus of an individual DexMA fiber that is used in this study? Based on the stretch distance (strain), Young's modulus, and diameter of a single fiber (Fig S1), can the authors make an estimate of the forces exerted? In addition, the migration modes may depend on the local "prestress" of the fibers. Indeed the way in which the fibers are interconnected modifies their relative tension. Is it the case?

Other points / questions:

How many cells are undergoing SSM-based migration among the entire population? I could hardly

find the statistics.

The relaxation mechanism does not appear very clear to me

The authors should provide data showing that the RGD concentration they used reflects the surface density of RDGs on the fibers (fig5). It may not be the case.

The author claim that "contact guidance may be a stronger effector of cell migration directionality than stiffness-induced polarization in fibrous ECM" (p6). This is a too strong statement. To make such a statement, the authors should provide data where cells could simultaneously experiment both cues, ie. fiber alignment and stiffness gradient, for instance. It is probably beyond the scope of the current study but this paragraph should be rephrased.

In the screen using additional cell lines and drugs, it would be helpful to show additional characteristics of the slingshots, for example the recoil distance (similar to Fig S6) and, in the case of cell lines, frequency.

Does the SSM migration mode have a preferred front/rear direction? In other words, if a cell is moving in one direction using continuous migration, does the SSM always move in the same direction?

In the main text it would be helpful to mention that the stiffness values reported for the samples represent the bulk elasticity (indented by ~1mm cylinder, orders of magnitude larger than the fiber spacing) and not the Young's modulus of a single fiber or the stiffness at length scale of a single cell (length scale similar to the fiber spacing).

Due to the discontinuous nature of SSM, it makes sense that standard deviation error bars can be large, it would be helpful to see some data represented to better see the population of data (for example, fast outliers for cells that frequently use SSM?).

I'm confused by Fig. 6E, it looks like experimental data but in the legend it states it is not.

Fig3 Legend. Second (f) should be (g).

Reviewer #1 (Remarks to the Author):

The manuscript by Wang et al. describes a purported new mode of migration that is termed “slingshot.” To reach this conclusion they utilize advanced fabrication methods to create fibrous environments where the mechanical rigidity, alignment, and biomolecular surface coating identity/concentration can be independently controlled. This is a very interesting experimental system, and provides a potentially rich platform for fundamental investigations into cellular processes that depend on these environmental parameters, such as cell migration. The most interesting results, in my view, are the possible optimal stiffness (Fig. 2E), the “slingshot” behavior, and its dependence on actomyosin contractility-adhesion. I will note that the “slingshot” behavior described here seems like the actomyosin-dependent “load and- fail” dynamics described by Chan and Odde (Science, 2008) for soft environments, although the present study goes further to relate these dynamics specifically to cell migration. The experiments are all well-executed and presented in a clear manner. Where I thought the manuscript was lacking was in its theoretical underpinnings, as the theorizing at the end of the manuscript did not have a mechanistic model with which to contextualize the findings. Given that recent studies have increasingly integrated such mathematical-computational modeling, the present manuscript was therefore not as sophisticated analyses (see reference below). Overall, this is an interesting study with important new observations on cell migration dynamics in state-of-the-art controlled environments, although there is room to improve by integration of mathematical-computational modeling.

Comments:

1. Lack of modeling makes the study not so cutting edge. For example, these recent studies from a number of different groups have all integrated mathematical-computational modeling that served not just for illustration purposes, but instead helped design experiments and interpret otherwise puzzling phenomena.

Stroka et al., Cell, 2014

Elosegui-Artola et al., Nat Materials, 2014

Chaudhuri et al., Nat Comm, 2015

Sunyer et al., Science, 2016

Elosegui-Artola et al., Nat Cell Biol, 2016

Bangasser et al., Nat Comm, 2017

Klank et al., Cell Reports, 2017

Oria et al., Nature, 2017

Weinberg et al., Biophys J, 2017

Mekhdjian et al., Mol Bio Cell, 2017

Estabridis et al., Ann Biomed Eng, 2018

Gong et al., PNAS, 2018

We agree with the reviewer that incorporating mathematical modeling would provide further insight and are working towards this direction, but we believe this significant body of experimental work is an important starting point that will inform the development of theoretical models in the future. Many of the publications listed above focus on dynamics at the length-scale of an adhesion, in particular modeling integrin-ECM engagement and actin flows using motor-clutch models where the ECM is modeled as an elastic or viscoelastic element. An important finding common to many of these efforts supports the idea of an optimal matrix stiffness that engenders maximal force transmission at the adhesion, and several of these works go on to connect this to protrusive or migratory behavior of cells. In our observation of a biphasic relationship between migration speed and matrix stiffness with a maximum noted at

intermediate stiffnesses, it is likely that a component of this response is due to integrin engagement and force transmission with focal adhesions. However, a key difference between these previous studies (largely performed on elastic or viscoelastic hydrogel surfaces) and the slingshot migration mechanism we describe here is the degree of matrix deformability observed, and furthermore the importance of this degree of stretch in directly influencing motion of the cell. In contrast to cells plated on typical hydrogels where material displacements are subcellular in length-scale (i.e. several microns), we note material displacements and resulting recoil distances that are larger than the cell (Supp. Fig. 9). Given the novelty of this observation, we chose to focus on this aspect that contributes to the biphasic response, but fully agree that adhesion/traction dynamics are another key component. As such, we have modified the text to clarify this point and made references to the above works that are relevant (pg. 6).

In contrast to hydrogels, discrete fibrous ECM with structural heterogeneity on the length-scale of the cell likely cannot be approximated as a mechanical continuum or spring. We suspect that directly applying the previously developed motor-clutch models will not fully capture the complexity of migration in these contexts. Given such structural heterogeneity which engenders pronounced variations in local mechanics¹, we anticipate integration of motor-clutch models with finite element approaches will be necessary. We are actively working with collaborators in this direction, with the hope that such an integrated model will provide new insights into the SSM mechanism. Why cells interconvert between continuous migration and SSM modes, how local stiffness changes during matrix reorganization and stretch, and how cells apply forces to the ECM are all exciting questions we hope to answer with such a model in the future.

2. Fig. 3E. Legend, part (g) is labeled (f)

We apologize for this error and have corrected the figure legend in the revised manuscript.

3. Fig. 5A. Why isn't the control (0 μ M) shown?

Dextran methacrylate was chosen for these studies due to its protein resistant nature, whereby cell adhesion to these matrices only occurs after active conjugation of a cell adhesive ligand. This feature enables us to control adhesive ligands without a confounding influence from adsorption of uncharacterized serum proteins. We have demonstrated this in the previous work introducing this material system². Previous studies with human mesenchymal stem cells indicate that cells do not spread when RGD is absent (reproduced below for Reviewer's ease). Similarly, dextran methacrylate substrates without cRGD (0 μ M) do not afford NIH3T3 spreading or migration, and thus we did not include this condition in the analysis.

Cell spreading on DexMA fiber networks as a function of RGD density after one day of culture. Cells have been stained for F-actin (green) and nuclei (blue) with phalloidin-Alexa 488 and Hoechst 33342, respectively; DexMA fibers have been labeled with rhodamine methacrylate (red). Cell spread area was quantified for $n > 37$ cells using a custom image analysis tool in Matlab. Scale bar: 50 μm .

4. Fig. 6B. The correlation with peak traction force and optimal stiffness is in Bangasser et al., Biophys J, 2013, Fig. 4B.

We have included this reference, as well as others that support an optimal stiffness for peak traction forces (see above comment). This work modeling substrate adhesion with a motor-clutch model importantly demonstrates how an optimal stiffness (amongst other parameters) can maximize traction forces that would drive cell protrusion and migration. As mentioned above, a key distinction between our observations and previous work using compliant hydrogel surfaces is the considerably higher matrix deformations we observe². While traction force generation captured by motor-clutch models is part of our observations, these existing models focused at adhesion scale phenomena do not capture the large-scale deformations and storage of elastic energy that we observe directly feed into cell motion. We have modified the text to clarify this point (pg. 6-7).

5. Need journal name: Drifka, C. R. et al. Periductal stromal collagen topology of pancreatic ductal adenocarcinoma differs from that of normal and chronic pancreatitis. 28, 1470–1480 (2015).

We have corrected this reference in the revised manuscript.

6. Aratyn -Schaus, Y., Oakes, P. W., Stricker, J., Winter, S. P. & Gardel, M. L. Preparation of Complaint Matrices for Quantifying Cellular Contraction. J. Vis. Exp. 1–6 (2010). doi:10.3791/2173. “Compliant” is misspelled

We have corrected this reference in the revised manuscript.

Reviewer #2 (Remarks to the Author):

This paper should be of interest to the readers, for both its methods and results. Synthetic electrospun fibers were suspended across silicone wells, to provide a 3D

fibrous environment that mimics the extracellular matrix. The approach avoids mechanical input from an underlying surface, thereby simplifying the interpretation. Methods were also developed to control the stiffness, orientation, and adhesiveness of the fiber. Cells embedded in these fibers showed an interesting mode of migration, referred to as sling shot migration, where surges of movement took place upon the release of tail, coupled to the recoil of fibers.

The study is not without significant weaknesses. First, the physiological significance of slingshot migration is questionable, as no such movement has been found in intravital imaging. The process requires a specific range of fiber density, configuration, orientation, and adhesiveness, which may be difficult to find in vivo.

Based on our studies, we would hypothesize that soft, elastic tissue environments that permit appreciable deformations under cell-generated traction forces while simultaneously building sufficient tension to trigger failure and elastic recoil would allow for this mode of migration. Towards addressing the question of physiologic relevance, we have additionally performed similar migration studies embedding NIH3T3 fibroblasts within fluorescently labeled 1.0 mg/ml type I collagen gels (Supp. Fig. 7, reproduced below for Reviewer's ease, Supp. Movie 7). In these settings, which we note are quite distinct from DexMA matrices in terms of fiber diameter, mechanics, and adhesive ligand, we also observe the occurrence of SSM. Following sudden rapid motion of the cell, an actin-containing portion of the cell is left attached to the relaxed substratum rearward to the cell's motion, similar to our studies with DexMA matrices. Additionally, we also qualitatively observed a condensation of collagen fibrils leading up to SSM followed by relaxation of the matrix upon recoil. Although these matrices were formed with disorganized fibrils, we would anticipate that inducing fibril alignment in these matrices would enhance SSM distances based on our studies comparing aligned vs. nonaligned DexMA matrices. These new data are now included in the revised manuscript with references in the main text (pg. 10).

Supplementary Figure 7: Slingshot migration within 3D type I collagen hydrogels. Select frames from confocal fluorescence time-lapse imaging (Supplemental Movie 7) of embedded NIH3T3-Lifeact-GFP cell migrating within Alexa555-succinyl ester labeled 1.0 mg/ml collagen 3D hydrogels (collagen-Alexa555 (cyan), Lifeact-GFP (magenta); scale bars: 50 μ m). Collagen gels were formed at 37° C (a) and 21° C (b). Arrows indicate Lifeact-GFP puncta left behind in the collagen following matrix recoil and slingshot migration.

We would also like to clarify that using DexMA matrices, we previously demonstrated that SSM can occur over a range of different matrix conditions including stiffnesses (Fig. 3d), in both highly aligned and randomly organized matrices (Fig. 3h), and with a variety of cell types (Fig. 6b). All of these parameters (and likely others we did not investigate here) influence characteristics of SSM, but this range of parameters supports the notion that this mode of migration can occur in diverse settings.

Given the short timescale of SSM (recoil occurring within a second), it is probable that these migration events would not be detected through intravital imaging which is performed at a much larger temporal resolution (minutes to hours). Despite this, previous intravital imaging studies have provided observations of matrix (and even collagen fiber) deformations during cell migratory events³. Given these important observations, we are actively pursuing a collaboration with a group specializing in intravital imaging in a breast cancer model to look for similar migration events in this *in vivo* context.

Lastly, we would like to draw an analogy to the observation of durotaxis, which was first described *in vitro*⁴ and to this day has not been convincingly observed intravitaly (to our knowledge), despite the general acceptance that durotaxis is an important migration cue and knowledge that gradients of stiffness exist throughout the body. We are hopeful that this study which highlights the influence of large-scale matrix stretch on cell motion is still an important contribution that could inform future studies and the field at large.

In addition, the percentage of cells undergoing slingshot migration appeared very low, around 1% (the authors should be more upfront about this; the text in line 132 simply stated “a subset of cells”).

We apologize for the confusion regarding the frequency of SSM events and have modified the text (pg. 7) and figures (Fig. 3 and 5) to clearly indicate the frequency of SSM events. The percentage of NIH3T3s that underwent slingshot migration on intermediate stiffness matrices is in fact close to 70% of all cells tracked during 6 hour long time-lapse imaging, meaning the majority of cells in this condition utilized SSM at least once (Figure 3d). To provide an additional measure of how often this mode of migration occurred, we have additionally quantified the % of the total imaging duration that cells spend in some phase of SSM. This duration of time predominantly involves matrix stretch as the recoil event occurs over one frame. At an intermediate matrix stiffness in aligned matrices, SSM comprises approximately 30% of the total tracked time (revised Fig. 3e). In other words, at an optimal stiffness nearly 1/3 of the average cell's total tracked time is spent stretching the matrix prior to a recoil event. In addition to providing these quantifications, we have clarified these measurements in the revised Methods (pg. 25).

The process also involved highly compressible properties of dextran methacrylate fibers that are not found in native ECM fibers.

In revision experiments, we have performed AFM-based three point bending of individual DexMA fibers over the range of crosslinking used in these studies. Young's modulus of single fibers is now included in Supplemental Figure 1a and vary from ~5 to ~15 MPa as a function of UV exposure time. These measurements are similar to reported values for fibrin fibers, elastin fibers, and fibronectin fibrils (depending on their stretch state)⁵⁻⁷.

Second, slingshot migration may be very similar if not identical to “retraction induced protrusion” as described decades ago by Chen (ref. 61), since pronounced tail retraction appeared to accompany the process. Evidence was weak that slingshot migration was driven by the recoil of elastic fibers, as proposed on line 192-193, rather than the retraction of tail as for retraction induced protrusion. Chen’s paper, cited amongst other references, needs to be addressed carefully. Major differences need to be presented in order to justify the present finding as a separate phenomenon.

Chen cultured chick fibroblasts on glass surfaces and observed considerable and abrupt motion of the cell’s trailing edge concurrent with a rupture event that left remnants of the cell attached to the substratum⁸. We view this observation as fully supportive of our claims regarding rupture/recoil, but with the key distinction between his observation and ours being the primary location of stretch. In his studies, he proposes mechanical tension within the cell’s actomyosin network underlies the elastic recoil upon rupture. This is supported by the fact that glass substrates do not deform under cell forces. In our model system, while actomyosin tension is certainly a central link in the matrix-cell-matrix system and the driver of force, it is the significant amount of stretch borne by the matrix that directly feeds into the motion of the cell upon rupture. We believe this is a critical distinction and highly relevant to those interested in understanding migration in soft, deformable tissue environments. We have amended the text to better clarify the distinction between Chen’s work and ours (pg. 10).

Specific Points:

- 1. Stiffness was measured with a bulk compression approach, which is likely very different from how a cell probes local fibers. Therefore the relevance of Young’s modulus as presented is questionable.**

To clarify, the bulk measurements that were performed were not compression tests, but rather involved tensile stretching of matrices by engagement of the surface with a cylindrical indenter. The downward motion of the indenter once engaged results in elongation of the area of material outside of the indenter’s circumference. Although we agree with the Reviewer that this does not capture local mechanical properties (which are likely highly heterogeneous given the discrete fibrous structure of these matrices), we do contend that the method reproduces the tensile nature of cell forces applied to the substrate. We have modified the main text to provide clarifications and caveats to the provided mechanical characterization (pgs. 5 and 14).

The Reviewer raises an excellent point that cells probe the material locally, so an additional metric reflecting the mechanical properties of the matrix could be the stiffness of individual fibers, which we have now provided (please see point above). However, we note that in hierarchically structured materials such as networks of fibers, the stiffness cells experience depends not only on the properties of individual fibers, but additionally their diameter, distribution, and orientation, how they are interconnected or crosslinked to each other, and proximity to rigid boundary conditions. The question of what length-scale of stiffness cells probe and furthermore the development of methods to measure such stiffnesses are major open challenges, and unfortunately beyond the scope of this study.

- 2. The presence of focal adhesions on synthetic fibers is not convincing (Supplemental Figure 4), which in turn affects the statements involving focal adhesions.**

We apologize for the poor quality images in the first submission of NIH3T3s constitutively expressing a paxillin-mRuby fusion tag protein. We agree that the high cytosolic signal and the absence of fluorescently labeled DexMA fibers (due to the shared channel between mRuby and

rhodamine) made identification of adhesions and general interpretation difficult. In the intervening months, we have created a new fusion tag construct with a brighter fluorophore, paxillin-EGFP, that enabled us to better visualize focal adhesions concurrently with rhodamine-labelled fibers. The enhanced contrast provided by this construct enabled better visualization of focal adhesions during trailing edge rupture. Please review Supp. Movie 6 and Supp. Fig. 5-6 (pgs. 7-8 below). Using this same construct, future work is planned to closely monitor focal adhesions dynamics during cell migration in these matrices.

3. **Also, the suggestion that “the recoil event is initiated by mechanical failure of trailing edge focal adhesions”, based on the appearance of residual focal adhesions and timescale, is not convincing.**

Towards providing a better visualization of these mechanical failure events, we have performed additional time-lapse experiments using NIH3T3s expressing Lifeact-GFP. Imaging at high resolution, we observe Lifeact-GFP puncta left tethered to the matrix rearward of the cell upon cell movement and matrix recoil (Supp. Fig. 5a (reproduced below), Supp. Movie 5).

Additionally, using the paxillin-EGFP construct mentioned above, we confirm the presence of paxillin-rich FAs within these cell remnants (Supp. Fig. 5b). Lastly, we performed time-lapse imaging capturing cells that left Lifeact-GFP remnants attached to the matrix and then immediately immunostained for vinculin, another well-accepted focal adhesion protein (Supp. Fig. 6a-b). This experiment indicates the ruptured actin-rich portion of the cytosol contains vinculin positive plaques. In sum, these experiments demonstrate that these actin-rich pieces of the cytosol left behind following cell recoil do contain paxillin and vinculin enriched puncta that are of the correct size to be FAs.

Supplementary Figure 5: Rupture of trailing edge concurrent with matrix recoil. (a) Select frames from confocal fluorescence time-lapse imaging shown in Supplemental Movie 5 of Lifeact-GFP expressing NIH3T3s within an aligned, intermediate stiffness matrix (matrix fibers (cyan), Lifeact-GFP (magenta); scale bar: 50 μ m). Arrows indicate Lifeact-GFP puncta tethered to the matrix with

matrix recoil. (b) Select frames from confocal fluorescence time-lapse imaging shown in Supplemental Movie 6 of Paxillin-GFP expressing NIH3T3s within an aligned, intermediate stiffness matrix (matrix fibers (cyan), cytoplasm (magenta), and paxillin (white), scale bar: 50 μm). Arrows highlight paxillin-rich puncta that remain tethered to the matrix within actin containing pieces of the cytoplasm that are separated from the cell upon matrix recoil. Roman numerals indicate paxillin-rich puncta associated across time frames.

Supplementary Figure 6: Confirmation of vinculin within ruptured trailing edge plaques that contain paxillin. (a) Select frames from confocal fluorescence time-lapse imaging of NIH3T3-Lifeact-GFP cells within an aligned, intermediate stiffness matrix (matrix fibers (cyan), Lifeact-GFP (magenta), and nuclei (yellow); scale bar: 50 μm). Matrix deformations are less visible due to low resolution imaging required for the sake of throughput. Arrow indicates Lifeact-GFP puncta tethered to matrix following matrix recoil. (b) Confocal fluorescence images of identical location as in (a), subsequently immunostained for vinculin directly on the microscope stage (matrix fibers (gray), Lifeact-GFP (magenta), nuclei (yellow), and vinculin (cyan); scale bar: 50 μm).

4. Clarify the duration for measuring the frequency of slingshot migration. It makes a major difference whether the 1% frequency was found over 10 minutes or 24 hours.

Please refer to our response to your point on this matter above. We have modified the main figures and text to clearly state the duration utilized to quantify the frequency of SSM events.

5. The effect of Calyculin on migration speed looks borderline (Figure 5d; $p \sim 0.05$). The two fold increase of slingshot frequency is more convincing, which then raises the question of why the average speed showed only a marginal difference.

The previously presented quantification migration speed for the calyculin A experiment incorporated all cells, including those that due to heightened contractility were unable to

respread following SSM and thus were subsequently nonmigratory. Inclusion of these non-migratory phases brought the resulting average speed of the population down. The inability of cells to spread in the presence of 1 nM calyculin A is supported by controls performed on tissue culture plastic, where freshly plated cells in the presence of this inhibitor similarly failed to spread or migrate. We have since re-performed our analysis only tracking cells for 1 hour following the addition of calyculin A, thereby excluding the subsequent period of time when the cells are non-migratory. These updated values for intermediate stiffness matrices are 43.53 $\mu\text{m/hr}$ in control and 62.01 $\mu\text{m/hr}$ upon 1 nM Calyculin A treatment, and are now included in revised Figure 5D.

- 6. The comparison between fiber-mediated contact guidance and stiffness-induced polarization (lines 114-115) is misleading, since the two involve different spatial and temporal frames. The former affects mainly migration persistence while the latter affects mainly the steering of protrusions.**

We fully agree with the reviewer and have removed this statement as comparing polarization is misleading across separate studies with different substrates and distinct bulk moduli.

Reviewer #3 (Remarks to the Author):

In this work, the authors electrospun DexMA fibers to generate well-controlled synthetic fibrous ECM and observed cell migration. On stiff matrices, cell migration is lower and on intermediate and soft matrices, cell migration is higher. Furthermore, the authors report a distinct mode of motility on softer matrices where the cells spread, contract the matrix, and then when a tension threshold is reached the rear adhesion will break and cell will "slingshot" forward using the front adhesions and recoil of the fibers. The authors further characterize this mode of migration and use various inhibitors and additional cell lines to reveal a more general correlation between traction forces and slingshot migration (SSM) frequency. I could not recommend the publication of the manuscript in Nature Communications at this stage. I have many major points that should be addressed.

Major points:

- 1. I found several claimed findings to be rather overstated. For instance, the so-called biphasic behavior of cell speed as a function of the stiffness should be clarified. Additional points between 0 and 7 kPa would be required to claim that there is indeed a biphasic behavior. I would be very cautious in the interpretation of the graph.**

We have performed additional studies to provide more data points and support the claim of a biphasic relationship between migration speed and stiffness in DexMA matrices. In the experiments in the initial submission (Figure 2e), we modulated UV exposure duration to tune DexMA crosslinking and resulting fiber and bulk stiffness. In revision experiments, we established an alternative method of modulating photoinitiator concentration to achieve two additional bulk modulus values between 0 and 7 kPa (Supp. Fig. 2a, reproduced below for Reviewer's ease). We performed migration studies with NIH3T3s at these additional stiffnesses and found graded increases in migration speed towards the optimal stiffness (Supp. Fig. 2b). Due to the distinct approach to crosslinking and the fact that this was a separate study run at a different time, we have elected not to superimpose these data points in Figure 2e of the main text, but have included them in the Supp. Fig. 2 and referenced them in the main text (pg. 6).

*Supplementary Figure 2: Bulk mechanical properties influence cell migration speeds. (a) Bulk mechanical testing of aligned matrices as a function of Irgacure photoinitiator (I2959) concentration with a constant UV exposure of 1 J/cm² (n>6 matrices/group). (b) NIH3T3 fibroblast migration speed as a function of Young's modulus in aligned matrices tuned by I2959 concentrations (a) (n>38 cells/stiffness). (c) Human foreskin fibroblast migration speed as a function of Young's modulus in aligned matrices with a constant 1.0 mg/ml I2959 but varying durations of UV exposure (as in Figure 1d) (n> 36 cells/stiffness). * indicates a statistically significant comparison with p<0.05.*

As a further demonstration of this biphasic response, we tracked human dermal fibroblasts across a range of bulk stiffnesses, and similarly find a biphasic response between migration speed and matrix stiffness (Supp. Fig. 2c). Interestingly, the optimal migration speed shifted to a higher stiffness than for NIH3T3s, matching previous reports performed on 2D polyacrylamide hydrogels where cell types with increased contractility have a higher modulus that results in optimal migration speed⁹. A reference to this result has been added to the text (pg. 6)

Another example is the discussion about FAs based on Sup Fig 4. The FA staining is far from being convincing. It remains difficult to advance that there is indeed a liberation of the entire FAs. The paxillin staining is localized everywhere within the entire cell.

Please refer to our response to Reviewer #2 comments #2-3 above, as these concerns were identical.

2. The porosity of the fibrillar matrix displays heterogeneities. How does it affect the measurements? Do the authors compare migration events over similar porosities for the different stiffness?

We have not closely examined how porosity influences migration in these settings, although we note that in our time-lapse imaging studies, very rarely were cells observed to be caught in pores. In these matrices, we anticipate porosity is influenced by fiber density and alignment, which was kept consistent across different stiffnesses by maintaining identical electrospinning conditions during the fabrication of substrates. Thus, in these experiments, all matrices begin with the same porosity. In response to your comment #6 below, we note that pore size is likely dynamic under cell induced deformations of the material. This is a challenging question that we are actively investigating, but given the rare observation of cell entrapment, we hypothesize pore size was not a major influence in these studies.

3. Slingshot frequencies should be expressed in units as a function of time, for example #events per time (otherwise I do not know common these events are; the authors

observe ~70% of cells over 8 hours imaging for intermediate stiffness in Fig 3D but that's the only information given).

Please refer to the response to Reviewer #2 (remarks to the author), who also indicated the frequency of SSM events are not sufficiently clearly presented. In brief, we have incorporated in the revised main figures additional quantification of the % of all imaged cells that undergo SSM as well as the % of total tracked duration spent in this mode of migration. We find that ~70% of NIH3T3s at the optimal matrix stiffness tested undergo SSM, and 30% of the tracked time of the average cell is spent in this migration mode.

4. The relaxation mechanism does not appear very clear to me. On intermediate stiffness (fig3), it seems that the fibers are bundled and stretched at the back of the cell after its movement. Could the authors clarify this point? How do they measure the relaxation and the matrix stretch? This point should be clarified as well.

In the majority of SSM events within aligned matrices, we noted condensation or compaction of matrix fibrils both rearward and forward of the center of the cell body. We anticipate the tensile nature of force generation by the cell and the predominance of anchoring points and active protrusive activity at the front and back of the cell would lead to recruitment of material at these same locations. After the cell has recoiled forward, the previously condensed matrix appears to quickly relax, although visually it appears to be an incomplete return to the initial configuration of the matrix. Shortly thereafter, a new round of fiber condensation can begin. Please examine Supp. Movie 2, as the still images do not fully convey the dynamics.

To demonstrate that matrix deformations involves tensile stretch of individual fibers, we randomly embedded fluorescent beads within DexMA fibers and tracked distances between paired beads within the same fiber during migration. To quantify recoil distances, we measured the distance traveled by the centroid of the cell nucleus from the end of the matrix stretch phase to the following frame after recoil. We have added details of this quantification to the revised Methods section (pg. 26). Although in other work we have noted these fibers behave elastically, viscoelastic behavior of the material would certainly influence relaxation and thereby alter SSM. Exploring the effect of viscoelasticity will be a focus of future work.

5. SSM seems to have very little effect on the overall migration speed of cells in intermediate stiffness matrices (Fig 2B vs 3F). Also, Fig. 3F is missing a bar. Treatment with Calyculin A doubles the SSM frequency and increases the distance, but the overall speed of migration has a very modest change (Fig. 5D). Taken together with the previous 2 comments, it seems this mode of migration has a small contribution towards the overall migration speed and does not explain why cells are migrating faster on soft and intermediate matrices than on stiff.

We apologize for the missing bar and have corrected this in the revised manuscript (Fig. 3g). Please note that quantification in Figure 3g was restricted to cells that interconverted between continuous and SSM (Fig. 3f). For this analysis, migration tracks of cells that underwent SSM were segregated into continuous, stretch, and recoil phases and speed was computed over these isolated periods. We argue this is the best possible comparison between continuous migration and SSM speeds, as it controls for a potential difference between cells that can undergo SSM and those who do not. A product of this approach is that periods of time where the cells stopped or changed direction were excluded from the analysis, resulting in an inflated continuous migration speed of 51.4 $\mu\text{m/hr}$, which as the Reviewer points out is very similar to the optimal migration speed on intermediate stiffness aligned matrices (Figure 2b, e). In contrast

to this analysis in Figure 3g, the quantification in Figure 2 is over a full 6 hour duration and is a composite measurement reflecting continuous migration, SSM events, as well as periods of pause or directional changes. We have reexamined the experiment in Figure 3g, analyzing the full tracks of the subset of cells that strictly continuously migrated and find a value of 40.0 ± 13.7 $\mu\text{m/hr}$. Although an imperfect comparison, this would imply that at the optimal intermediate stiffness, the contribution of SSM to the average migration speed is ~ 10 $\mu\text{m/hr}$. We agree with the Reviewer that SSM is not the full explanation for differences in speed as a function of stiffness, but only one contribution that we focus on given its novelty. We have modified the text to clarify this point (pg. 7), as well as the description of the analysis in Figure 3g.

Regarding the point about calyculin A, the same question was raised in Reviewer #2 comment #5. Please see our response above detailing the updated analysis in the revised manuscript.

6. One simple explanation for cells moving much slower in stiff 3D fibers is that the cell is unable to deform the fibers to make space for the cell to squeeze through, whereas compliant fibers would allow for this. Have the authors checked whether the cells are on top of the fiber matrix (2D) or if the cells are fully embedded within the fibers (3D)? (Fig S1C has a confocal image and it appears the cell is fully embedded, but it's not clear how general this is).

Similarly to cell derived matrix¹⁰, cells are initially seeded on top of DexMA fibrous matrices and with time can infiltrate and become fully embedded within 3D. In our experiments, cells are seeded and cultured for 6 hours to enable infiltration into the fibrous matrix prior to time-lapse imaging. We have performed additional quantification of the depth of NIH3T3s, finding that the vast majority of cells are fully embedded within the matrix (meaning there are DexMA fibers both above and below the cell body). This quantification has been provided in Supp. Fig. 1d-f and referenced in the main text (pg. 6).

The reviewer suggests that rigidity of matrix pores could influence migration speeds, however in all of our time-lapse data sets, cells caught in pores was observed rarely. We have not quantified pore size in these matrices given the difficulties implementing methods such as mercury porosimetry in soft materials, but we anticipate the average pore size is above the critical $\sim 2\text{-}3$ μm threshold reported by Friedl and colleagues. This is supported by the observation that cells rapidly infiltrate matrices of varying stiffness and fiber alignment within the first few hours after seeding. While we did not see this effect in these studies, we do agree that the topic is of interest (and this is supported by high profile works from the Konstantopoulos and Lammerding groups, amongst others). One counter-argument to the notion suggested by the reviewer is that highly deformable matrices result in condensation of fibrils local to the cell that would effectively reduce pore dimensions. Ongoing collaborative work is focused on examining these potential conflicting effects in more detail.

7. The authors observe stretching of individual fibers by tracking bead positions as fiducial markers. In a 3D collagen system, the Young's modulus of a fibril can be $\sim \text{GPa}$ thus it is unlikely this would be the case (would this affect SSM?). What is the Young's modulus of an individual DexMA fiber that is used in this study? Based on the stretch distance (strain), Young's modulus, and diameter of a single fiber (Fig S1), can the authors make an estimate of the forces exerted? In addition, the migration modes may depend on the local "prestress" of the fibers. Indeed the way in which the fibers are interconnected modifies their relative tension. Is it the case?

The reviewer correctly points out that the Young's modulus of individual type I collagen fibers is in the GPa range, with values ranging from ~0.1 to 10 GPa (variability of reported values is likely influenced by the testing modality, the source of the collagen, and the method used to isolate the collagen fibril)^{6,11,12}. However, when assembled hierarchically into a matrix, the resulting stiffness that cells experience is far lower (in the kPa range) and a function of the porosity of the matrix, the diameter of the fibrils, the interconnections between fibrils, and the proximity to rigid boundary conditions that constrain the gel, amongst other parameters. As one illustration to this point, Doyle et al. used AFM to measure the mechanics of individual collagen fibrils within a formed gel and reported values in the range of ~1-10 kPa¹³.

In revision experiments, we have established in the lab the AFM three point bending method to quantify the Young's modulus of individual DexMA fibers used in these studies. These values are presented in Supplemental Figure 1a and vary from ~5 to ~15 MPa over the range of DexMA crosslinking tuned by UV exposure employed in these studies. These measurements are similar to reported values for fibrin fibers, elastin fibers, and fibronectin fibrils (depending on their stretch state)⁵⁻⁷, although as stated above, the mechanics that cells experience is a composite of numerous properties in addition to the Young's modulus of individual fibers.

Given these measurements, we can estimate the force imparted on a single fiber. The largest stretch ratio we observed was ~1.5 (50% strain of the fiber relative to unloaded) in fiber matrices of intermediate stiffness. Assuming more representative strains of 10% below this extreme, an average fiber diameter of 1.12 μm (Supp. Fig. 1b), a determined single fiber Young's modulus for these matrices of ~9.4 MPa, and finally that the fiber is a linear elastic material, we could estimate the force in an assumed fiber as:

$$\sigma = F/A = E*\epsilon, \text{ where } A = 0.99 \mu\text{m}^2 \text{ is the cross-sectional area of a } 1.12 \mu\text{m fiber}$$
$$F = 9.4 \times 10^6 \text{ N/m}^2 \times 0.1 \text{ (strain)} \times 0.99 \times 10^{-12} \text{ m}^2 = 0.93 \mu\text{N}$$

Although this value is of the correct order of magnitude for reported cell generated forces, we note that an actual quantification of cell forces in settings composed of discrete fibers is incredibly challenging and beyond the scope of the work at hand. This is evident in the number of assumptions made above to approximate a force, the heterogeneity of the material (i.e. geometric and mechanical properties vary considerably from fiber to fiber), and the need for additional mechanical characterization including viscous and nonlinear behaviors. We are currently working with collaborators to develop finite element modeling approaches and improved mechanical testing methods, with the long-term goal of being able to quantify cell traction forces in these settings. Such methods will hopefully enable us to better predict the local mechanical conditions that enable slingshot migration, one aspect of which is slack or prestress in the material as the Reviewer points out.

Other points / questions:

8. How many cells are undergoing SSM-based migration among the entire population? I could hardly find the statistics.

Please refer to the response to Reviewer #2 (remarks to the author) above who raised an identical concern, as well as your comment #3 above.

9. The relaxation mechanism does not appear very clear to me

We assume the Reviewer is referring to the recoil events, which was raised by Reviewer #2 in comments #2-3 above. Please see the more detailed response above, but in brief we have

developed a new paxillin reporter construct and performed additional high resolution time-lapse imaging along with immunostaining to better describe recoil events and mechanical failure at the trailing edge of the cell during SSM.

10. The authors should provide data showing that the RGD concentration they used reflects the surface density of RDGs on the fibers (fig5). It may not be the case.

The reviewer brings up a common challenge to synthetic biomaterial approaches involving the active conjugation of adhesive peptides. Commonly accepted methods for determination of an absolute density of RGD molecules have not been established. NMR or FTIR prove difficult due to the small concentration of RGD compared to the base material (DexMA in these studies), and NMR further requires re-solubilization of the material. In previous work² (data reproduced below for Reviewer's ease), we conjugated a fluorophore to RGD and quantified fluorescence intensity to demonstrate that our input concentrations reflected the resulting material's adhesivity that cells encountered. Importantly, we showed that the tuned stiffness of DexMA fibers did not impact the apparent coupling efficiency, and that indeed we could modulate the coupled density of RGD molecules, however this method is not without limitations. In the studies presented here, we screened a range of cRGD concentrations and selected a concentration deemed to be saturating based on steady state cell spreading. This data is included in Supp. Fig. 1c.

Relative quantification of RGD density coupled to DexMA fiber networks of different stiffnesses. a, Confocal maximum projections of DexMA fibers coupled with RGD (left) and FITC-RGD (middle) via a Michael type addition. Additional samples were incubated in identical conditions with FITC to detect the background fluorescence due to diffusion and passive adhesion (right). b, Soft and stiff fiber networks coupled with a range of FITC-RGD concentrations, demonstrating RGD coupling is independent of network stiffness. c, Quantification of fluorescence intensity from confocal stacks. No statistically significant differences were identified when comparing soft and stiff at a given RGD concentration (mean \pm s.d., $n \geq 5$ ROI, significance set at $P < 0.05$).

11. They author claim that “contact guidance may be a stronger effector of cell migration directionality than stiffness-induced polarization in fibrous ECM” (p6). This is a too strong statement. To make such a statement, the authors should provide data where cells could simultaneously experiment both cues, ie. fiber alignment and stiffness gradient, for instance. It is probably beyond the scope of the current study but this paragraph should be rephrased.

We agree with the reviewer that our current studies do not provide sufficient conditions to support this claim. As such, we have removed this claim.

12. In the screen using additional cell lines and drugs, it would be helpful to show additional characteristics of the slingshots, for example the recoil distance (similar to Fig S6) and, in the case of cell lines, frequency.

In our contractility perturbation studies, inhibition of contractility with 30 μM blebbistatin and 25 μM Y27362 resulted in a complete abrogation of slingshot events (Fig. 5b-c), rendering further characterization of slingshot impossible. To address this, we have performed additional experiments utilizing lower dosages of blebbistatin (5 μM), which resulted in a partial decrease of contractility as determined by TFM (Supp. Fig. 8b, see below). At this intermediate level of contractility, we note a decrease in the fraction of the total cell population that undergoes SSM as well as a decrease in the percentage of time spent in SSM mode (Supp. Fig. 8d, see below).

*Supplementary Figure 8: (b) Peak traction stress of NIH3T3s as a function of blebbistatin concentration measured via traction force microscopy on 7.9 kPa PAAm hydrogels ($n > 25$ cells/condition). (c-d) Migration speed, SSM population (quantified over a 6-hour duration) and SSM duration of NIH3T3s treated with an intermediate blebbistatin dosage (5 μM) on aligned, intermediate stiffness matrices ($n > 84$ cells/group (c), $n = 3$ fields of view; field of view = 10 cells (d)). * indicates a statistically significant comparison with $p < 0.05$.*

Additionally, as requested, we have determined recoil distances for the 5 μM blebbistatin condition and added this data to Supp. Fig. 9 (reproduced below for Reviewer’s ease).

Supplementary Figure 9: Matrix stretch duration and recoil distance is dependent on intracellular contractility. Recoil distance (net translocation of cell) as a function of duration spent stretching the matrix. Dashed lines indicate corresponding linear correlations with indicated R^2 and p -values.

Lastly, for the cell screening experiment, we now provide the SSM duration as requested in Supp. Fig. 10 (reproduced below for Reviewer's ease).

Supplementary Figure 10: Optimal stiffness for slingshot migration varies by cell type. % of total imaging duration cells spent in some phase of matrix stretch and recoil (SSM duration) as a function of Young's modulus for a variety of cell types. Dashed lines connect data points for each cell type.

13. Does the SSM migration mode have a preferred front/rear direction? In other words, if a cell is moving in one direction using continuous migration, does the SSM always move in the same direction?

We found cells with a single defined leading edge resulted in recoil preferentially towards the direction of continuous migration and the leading edge, while cells with bidirectional extensions showed in no preferential direction with respect to the direction of previous migration. This data is included in Supp. Fig. 3b (reproduced below for Reviewer's ease).

*Supplementary Figure 3b: Percentage of cells employing SSM that recoil in the same direction (forward) or in the reverse direction (backward) with respect to its direction of continuous migration prior to SSM. Cells were parsed into two categories (leading edge and bidirectional) based on the morphology of cell protrusions during matrix stretch. * indicates a statistically significant comparison with $p < 0.05$.*

14. In the main text it would be helpful to mention that the stiffness values reported for the samples represent the bulk elasticity (indented by ~1mm cylinder, orders of magnitude larger than the fiber spacing) and not the Young's modulus of a single fiber or the stiffness at length scale of a single cell (length scale similar to the fiber spacing).

To address Reviewer #2 comment #1 as well as your comment above (#7), we have performed AFM three point bending tests to determine average Young's moduli of the individual DexMA fibers as a function of crosslinking. These values are now provided in Supp. Fig. 1a. Further, we have modified the main text to more clearly describe our mechanical measurements, clarify these measurements represent a bulk elasticity, and alert the reader to the corresponding single fiber values (pg. 5). Please see the responses above for additional comments on the complexity of mechanics in fibrous settings.

15. Due to the discontinuous nature of SSM, it makes sense that standard deviation error bars can be large, it would be helpful to see some data represented to better see the population of data (for example, fast outliers for cells that frequently use SSM?).

For all migration speed quantifications, we have now included all data points superimposed upon bars representing the mean so that readers can see the spread to the data. We note that the standard deviation for these measurements did not appear to be any larger than the intrinsic spread to other biological measurements. For example, the quantification of average spread area (Supp. Fig. 1c) also revealed a standard deviation of approximately 30% of the mean. We interpret this to be an outcome of the fact that migration speed was quantified over a significant 6 hour duration.

16. I'm confused by Fig. 6E, it looks like experimental data but in the legend it states it is not.

We apologize for the confusion. Figure 6E is composed of theoretical data and jagged lines were chosen to illustrate the traction force fluctuations that we anticipate exist based on our time-lapse imaging and others' work^{14,15}. We have sought to clarify the nature of this plot in the revised figure legend (pg. 22).

17. Fig3 Legend. Second (f) should be (g).

We apologize for this error and have corrected the figure legend.

References:

1. Beroz, F. *et al.* Physical limits to biomechanical sensing in disordered fibre networks. *Nat. Commun.* **8**, 16096 (2017).
2. Baker, B. M. *et al.* Cell-mediated fibre recruitment drives extracellular matrix mechanosensing in engineered fibrillar microenvironments. *Nat. Mater.* **14**, 1262–1268 (2015).
3. Perentes, J. Y. *et al.* In vivo imaging of extracellular matrix remodeling by tumor-associated fibroblasts. *Nat. Methods* **6**, 143–145 (2009).
4. Lo, C. M., Wang, H. B., Dembo, M. & Wang, Y. L. Cell movement is guided by the rigidity of the substrate. *Biophys. J.* **79**, 144–152 (2000).
5. Collet, J.-P., Shuman, H., Ledger, R. E., Lee, S. & Weisel, J. W. The elasticity of an individual fibrin fiber in a clot. *Proc. Natl. Acad. Sci.* **102**, 9133–9137 (2005).
6. Guthold, M. *et al.* A Comparison of the Mechanical and Structural Properties of Fibrin Fibers with Other Protein Fibers. *Cell Biochem. Biophys.* **49**, 165–181 (2007).
7. Klotzsch, E. *et al.* Fibronectin forms the most extensible biological fibers displaying switchable force-exposed cryptic binding sites. *Proc. Natl. Acad. Sci.* **106**, 18267–18272 (2009).
8. Chen, W.-T. Mechanism of the trailing edge during fibroblast movement. *J. Cell Biol.* **90**, 187–200 (1981).
9. Bangasser, B. L., Rosenfeld, S. S. & Odde, D. J. Determinants of maximal force transmission in a motor-clutch model of cell traction in a compliant microenvironment. *Biophys. J.* **105**, 581–592 (2013).
10. Hakkinen, K. M., Harunaga, J. S., Doyle, A. D. & Yamada, K. M. Direct comparisons of the morphology, migration, cell adhesions, and actin cytoskeleton of fibroblasts in four different three-dimensional extracellular matrices. *Tissue Eng. Part A* **17**, 713–24 (2011).
11. An, K. N., Sun, Y. L. & Luo, Z. P. Flexibility of type I collagen and mechanical property of connective tissue. *Biorheology* **41**, 239–46 (2004).
12. van der Rijt, J. A. J., van der Werf, K. O., Bennink, M. L., Dijkstra, P. J. & Feijen, J. Micromechanical Testing of Individual Collagen Fibrils. *Macromol. Biosci.* **6**, 697–702 (2006).
13. Doyle, A. D., Carvajal, N., Jin, A., Matsumoto, K. & Yamada, K. M. Local 3D matrix microenvironment regulates cell migration through spatiotemporal dynamics of contractility-dependent adhesions. *Nat. Commun.* **6**, (2015).
14. Chan, C. E. & Odde, D. J. Traction Dynamics of Filopodia on Compliant Substrates. *Science* **322**, 1687–1691 (2008).
15. Plotnikov, S. V., Pasapera, A. M., Sabass, B. & Waterman, C. M. Force fluctuations within focal adhesions mediate ECM-rigidity sensing to guide directed cell migration. *Cell* **151**, 1513–1527 (2012).

REVIEWERS' COMMENTS:

Reviewer #1 (Remarks to the Author):

The main concern from the previous review was the lack of a theoretical component. The authors responded that it is the experimental observation of the degree of matrix deformability and its direct influence on cell migration that is novel, a point which I agreed with in my initial review. In their rebuttal, the authors make a good point that the magnitude of the load-and-fail displacements is much larger than previously reported (e.g. in Chan and Odde, 2008).

However, the main concern about mechanism remains: is the slingshot dynamic basically a manifestation of the physics of the motor-clutch model, played out in this experimental system? Or is some new cellular mechanics at play? The authors in their rebuttal seem to equivocate, contending that they suspect the motor-clutch mechanism is not at play, or at least would require a new parameterization of the environment if not the cell. First, I will note that such parameterization of the motor-clutch model to account for fibrous geometries and mechanical anisotropy has already been reported by Estabridis et al. (2018), a study mentioned in my previous review but not mentioned in the rebuttal and not cited in revision. More importantly, the suspicion that the authors have with regard to mechanism is not backed by a scientific justification, either in the rebuttal or the revised manuscript. Thus, the revised manuscript, as it stands, comes across as observational, albeit with an interesting observation.

Therefore, to bring the study closer to the standard in the field (i.e. one that includes explicit biophysical modeling), the discussion section of the manuscript should include some discussion of mechanism, i.e. slingshot motion via the motor-clutch model, or an alternative model if that is favored by the authors for reasons that can be articulated. If the authors are not able to provide biophysical modeling and simulation results to support or refute hypothetical mechanisms, then they need to articulate some mechanism able to explain their novel observations.

Reviewer #2 (Remarks to the Author):

The authors have made serious efforts addressing previous concerns and answering most questions. My remaining quibble is that the relationship to retraction induced migration could have been better discussed than the brief dismissive statement on p.10. It remains quite possible that the two phenomena represent the same cellular event with different manifestations in environments of different stiffness. The difference as stated on p.10 pertains only to a disagreement with the hypothesis proposed by Chen. As the previous report was published 30+ years ago without the current technology, I don't see the value of the current study being diminished by a more thoughtful discussion of the relationship. Instead, it may only place the results on a firm historical foundation and enhance the appreciation.

Reviewer #3 (Remarks to the Author):

I am satisfied with the author's responses to my previous criticisms. They have worked very hard both to clarify their statements and to generate new data to support their interpretations. I now support the publication of this paper.

REVIEWERS' COMMENTS:

Reviewer #1 (Remarks to the Author):

The main concern from the previous review was the lack of a theoretical component. The authors responded that it is the experimental observation of the degree of matrix deformability and its direct influence on cell migration that is novel, a point which I agreed with in my initial review. In their rebuttal, the authors make a good point that the magnitude of the load-and-fail displacements is much larger than previously reported (e.g. in Chan and Odde, 2008).

However, the main concern about mechanism remains: is the slingshot dynamic basically a manifestation of the physics of the motor-clutch model, played out in this experimental system? Or is some new cellular mechanics at play? The authors in their rebuttal seem to equivocate, contending that they suspect the motor-clutch mechanism is not at play, or at least would require a new parameterization of the environment if not the cell. First, I will note that such parameterization of the motor-clutch model to account for fibrous geometries and mechanical anisotropy has already been reported by Estabridis et al. (2018), a study mentioned in my previous review but not mentioned in the rebuttal and not cited in revision. More importantly, the suspicion that the authors have with regard to mechanism is not backed by a scientific justification, either in the rebuttal or the revised manuscript. Thus, the revised manuscript, as it stands, comes across as observational, albeit with an interesting observation.

Therefore, to bring the study closer to the standard in the field (i.e. one that includes explicit biophysical modeling), the discussion section of the manuscript should include some discussion of mechanism, i.e. slingshot motion via the motor-clutch model, or an alternative model if that is favored by the authors for reasons that can be articulated. If the authors are not able to provide biophysical modeling and simulation results to support or refute hypothetical mechanisms, then they need to articulate some mechanism able to explain their novel observations.

We agree that previously described motor-clutch models are relevant to this work and would accurately describe adhesion dynamics and actomyosin force transmission to the ECM that is central to our observations. These models elegantly provide insight into the molecular events that contribute to adhesion growth, contractility, and migration, and recent versions have considered multiple adhesions spatially distributed across the cell as well as substrates that are fibrous in nature (Estabridis et al., 2018), an important reference which we have now incorporated into the revised discussion. While these models demonstrate how matrix mechanics influences adhesion behavior to influence force transmission and cell migration, to our knowledge these models do not incorporate cell translocation due to large-scale matrix stretch and subsequent recoil. This behavior likely arises from the unique material properties of these deformable fibrous matrices and we fully believe that future iterations and reparameterization of such motor-clutch models could capture such behavior. Additionally, incorporating heterogeneity into these models to reflect the local variability in mechanics and structure intrinsic to these and natural tissue settings could be an important next step towards understanding the switching behavior we observed in these studies. We have modified our

discussion to clearly associate the motor-clutch model with our observations and articulate our perspective on important future steps to improve these models.

“The growth and decay of focal adhesions and resulting dynamics of force transmission to the ECM have previously been described using motor-clutch models, which yield a qualitatively similar biphasic response of migration speed to matrix stiffness that is likely at play in our studies¹⁻⁵. Such models recently have been extended to fibrous matrices⁶, and further adopting these models to incorporate matrix heterogeneity and cell translocation due to large-scale matrix stretch may provide additional insights into the observations described here.”

Reviewer #2 (Remarks to the Author):

The authors have made serious efforts addressing previous concerns and answering most questions. My remaining quibble is that the relationship to retraction induced migration could have been better discussed than the brief dismissive statement on p.10. It remains quite possible that the two phenomena represent the same cellular event with different manifestations in environments of different stiffness. The difference as stated on p.10 pertains only to a disagreement with the hypothesis proposed by Chen. As the previous report was published 30+ years ago without the current technology, I don't see the value of the current study being diminished by a more thoughtful discussion of the relationship. Instead, it may only place the results on a firm historical foundation and enhance the appreciation.

We apologize that the reference to Chen's important work came across as dismissive, as this was not our intention. We agree with the reviewer that the two phenomena may be linked in terms of the underlying generation of actomyosin tension, played out differently due to the notable distinctions in deformability and elasticity of the underlying substrate in our studies compared to his. We have extended the discussion on his work, as follows:

“These abrupt mechanical failure events bare resemblance to retraction-induced migration described in chick fibroblasts cultured on rigid glass substrates by Chen several decades ago⁷, where tension in the cytoskeleton was hypothesized to cause elastic recoil of the cell body upon trailing edge detachment. In contrast, however, our studies employing deformable elastic substrates suggest tension stored additionally in matrix fibrils can induce substrate recoil and simultaneous cell translocation upon rupture at the trailing edge.”

Reviewer #3 (Remarks to the Author):

I am satisfied with the author's responses to my previous criticisms. They have worked very hard both to clarify their statements and to generate new data to support their interpretations. I now support the publication of this paper.

References:

- 1 B. L. Bangasser, G. A. Shamsan, C. E. Chan, K. N. Opoku, E. Tüzel, B. W. Schlichtmann, J. A. Kasim, B. J. Fuller, B. R. McCullough, S. S. Rosenfeld and D. J. Odde, *Nat. Commun.*, 2017, **8**, 15313.
- 2 R. L. Klank, S. A. Decker Grunke, B. L. Bangasser, C. L. Forster, M. A. Price, T. J. Odde, K. S. SantaCruz, S. S. Rosenfeld, P. Canoll, E. A. Turley, J. B. McCarthy, J. R. Ohlfest and D. J. Odde, *Cell Rep.*, 2017, **18**, 23–31.
- 3 F. Qu, Q. Li, X. Wang, X. Cao, M. H. Zgonis, J. L. Esterhai, V. B. Shenoy, L. Han and R. L. Mauck, *Sci. Rep.*, 2018, **8**, 3295.
- 4 A. Elosegui-Artola, R. Oria, Y. Chen, A. Kosmalska, C. Pérez-González, N. Castro, C. Zhu, X. Trepát and P. Roca-Cusachs, *Nat. Cell Biol.*, 2016, **18**, 540–548.
- 5 R. Oria, T. Wiegand, J. Escribano, A. Elosegui-Artola, J. J. Uriarte, C. Moreno-Pulido, I. Platzman, P. Delcanale, L. Albertazzi, D. Navajas, X. Trepát, J. M. García-Aznar, E. A. Cavalcanti-Adam and P. Roca-Cusachs, *Nature*, 2017, **552**, 219–224.
- 6 H. M. Estabridis, A. Jana, A. Nain and D. J. Odde, *Ann. Biomed. Eng.*, 2018, **46**, 392–403.
- 7 W.-T. Chen, *J. Cell Biol.*, 1981, **90**, 187–200.